# A Circular Argument:
# Does RoPE *need* to be Equivariant for Vision?

**Chase van de Geijn**[1,†], **Timo Lüddecke**[1], **Polina Turishcheva**[1]    **Alexander S. Ecker**[1,2,*]

[1]Institute of Computer Science and Campus Institute Data Science, University of Göttingen
[2]Max Planck Institute for Dynamics and Self-Organization, Göttingen, Germany
[†]chase.geijn@uni-goettingen.de
[*]ecker@cs.uni-goettingen.de

## Abstract

Rotary Positional Encodings (RoPE) have emerged as a highly effective technique for one-dimensional sequences in Natural Language Processing spurring recent progress towards generalizing RoPE to higher-dimensional data such as images and videos. The success of RoPE has been thought to be due to its positional equivariance, i.e. its status as a *relative* positional encoding. In this paper, we mathematically show RoPE to be one of the most general solutions for equivariant positional embedding in one-dimensional data. Moreover, we show Mixed RoPE to be the analogously general solution for $M$-dimensional data, if we require commutative generators – a property necessary for RoPE's equivariance. However, we question whether strict equivariance plays a large role in RoPE's performance. We propose Spherical RoPE, a method analogous to Mixed RoPE, but assumes non-commutative generators. Empirically, we find Spherical RoPE to have the equivalent or better learning behavior compared to its equivariant analogues. This suggests that relative positional embeddings are not as important as is commonly believed, at least within computer vision. We expect this discovery to facilitate future work in positional encodings for vision that can be faster and generalize better by removing the preconception that they must be relative.

## 1   Introduction

Deep learning is in the age of transformers [73]. At their core, transformers are built on attention [4, 61], which is a permutation-invariant operation [74], making them agnostic to word or token position within a corpus. To break this symmetry, tokens must be modified with position embeddings [25, 83]. Recently, Rotational Positional Encodings (RoPE) [69] have gained popularity, touting an emphasis on the *relative* position between two tokens rather than their absolute positions [22, 26, 43, 46]. However, some of the original claims of RoPE have been called into question leading to confusion as to *why* it works: Su et al. [69] claimed the attention scores to decay with distance between tokens. This was found to be true only for attention with the same query and key [5]. Moreover, transformers with causal masking have been shown to require no positional encodings to be capable of recovering absolute position [24], making RoPE's relative (shift-equivariant) claim questionable. However, many new methods continue to be motivated by RoPE's benefit from shift-equivariance [27, 60, 85] . To guide future research in positional encodings, it is important to discover whether shift-equivariance truly makes RoPE successful and needs to be preserved when extending it.

Both transformer and RoPE were originally designed for one-dimensional sequences such as language. RoPE encodes position by pairing dimensions within the query and key vectors within a transformer and rotating the paired dimensions. Transformers have become the current staple across all AI fields [17, 29, 46, 47, 63]. Naturally, RoPE's recent popularity in NLP has also spread to Vision

Transformers (ViT), where the data is two- or three-dimensional, corresponding to images and videos. *How* to extend RoPE to other modalities is nontrivial and assumptions must be made to maintain equivariance [44, 60]. The most commonly used approach for extending RoPE to ViTs is through Axial RoPE [15, 26, 77], partitioning the embedding dimensions into dimensions rotated independently either by the horizontal or the vertical position of the tokens. However, this approach does not allow for diagonal attention patterns where horizontal and vertical information "mix", which have been hypothesized to enhance generalization; consequently, learned Mixed RoPE was proposed [26]. Even more recently, LieRE [52] generalized Mixed RoPE from pair-wise rotations to higher dimensional rotations using learned skew-symmetric Lie algebras. If one defines rotations to be special orthogonal transformation, LieRE is the most general form of rotation encoding. However, while general, LieRE does not guarantee equivariance.

In this paper, we investigate the relationships across these different forms of positional encoding. In Section 3, we mathematically show RoPE with parameterized rotation speeds to be equivalent to LieRE for one-dimensional data. When the number of positional dimensions is higher dimensional, LieRE is not guaranteed to be equivariant unless constraints are placed on the Lie algebras. Using this insight, we derive Axial RoPE by imposing a "mutual exclusivity" constraint on the eigenvalues of LieRE's generators. Further, we will show that if one loosens this constraint – requiring the Lie algebra to be commutative between the generator – then one arrives at Mixed RoPE. To be a relative encoding, this commutativity property is necessary [44, 60], thus making Mixed RoPE the most general form of LieRE which maintains equivariance. However, it has been noted that requiring the positional embedding to be relative is an inductive bias whose necessity to RoPE's success is unclear [1, 5, 24].

The perceived necessity of equivariance has led to a circular argument where positional embeddings are assumed to perform well because they are relative, and all new embeddings must be relative because relative embeddings perform well. To break this cycle, we believe that it is imperative to establish the importance of equivariance embeddings for multi-dimensional RoPE. In Section 4, we propose alternative methods to establish a cause-effect experiment to evaluate whether equivariances is a predominant contributor to RoPE's faster training dynamics and generalization. To this end, we propose Spherical RoPE which takes a non-commutative assumption, thus breaking equivariance, and Uniform RoPE, which maintains equivariance, but has only a single shared rotation speed.

In Section 5, we find that Spherical RoPE has the same training behaviors as its equivariant analogues and we find that Uniform RoPE outperforms the standard learned encodings, while performing worse than other RoPE methods. We conclude that our evidence suggests that the performance of RoPE over traditional embeddings is not explained by equivariance.

## 2   Background

In this section, we review concepts and notation from previous work on rotary positional embeddings. We introduce the methods in both historical and progressively general order which we will use to prove in Section 3 that Mixed RoPE is the most general $M$-D rotary embedding with equivariance. For a broader literature review on positional embeddings see Appendix C. For a compact overview of symbols, see Appendix D.

### 2.1   Attention

We use the standard attention mechanism from Vaswani et al. [73], given by

$$\mathbf{Z} = \text{Attention}(\mathbf{Q}, \mathbf{K}, \mathbf{V}) = \text{softmax}\left(\frac{\mathbf{Q}\mathbf{K}^\top}{\sqrt{d_k}}\right)\mathbf{V}. \tag{1}$$

We consider only single-headed attention to simplify notation, so here $\mathbf{Q}, \mathbf{K}$, and $\mathbf{V}$ are elements of $\mathbb{R}^{T \times N}$, where $T$ is the number of tokens and $N$ is the network's latent dimension. We will primarily use index notation, where the above equation is expressed as, $\mathbf{z}_i = \sum_{j=1}^{\top} a(\mathbf{q}_i, \mathbf{k}_j)\mathbf{v}_j$. We define the attention mechanism, $a(\mathbf{q}_i, \mathbf{k}_j)$, as

$$a(\mathbf{q}_i, \mathbf{k}_j) = \frac{e^{\alpha(\mathbf{q}_i, \mathbf{k}_j)}}{\sum_{j=1}^{T} e^{\alpha(\mathbf{q}_i, \mathbf{k}_j')}}, \tag{2}$$

where what we refer to as the *attention score* is given by

$$\alpha(\mathbf{q}, \mathbf{k}) = \mathbf{q}^\top \mathbf{k}. \tag{3}$$

This formulation of attention is equivariant to permutations of the token order. To break this symmetry, the position of the tokens must be "encoded" into the attention scores. Thus, we re-express the attention score as a function of the content of the query token $\mathbf{x}_i \in \mathbb{R}^N$ and key token $\mathbf{x}_j \in \mathbb{R}^N$, and their positions $p_i, p_j \in \mathbb{R}$,

$$\alpha_{ij} := \alpha(\mathbf{q}_i, \mathbf{k}_j) := \alpha((\mathbf{x}_i, p_i), (\mathbf{x}_j, p_j)) := \alpha(\mathbf{x}_i, \mathbf{x}_j, p_i, p_j). \tag{4}$$

Throughout this paper, we will abuse the notation of $\alpha$ and use these expressions interchangeably for ease of notation. If the position affects the query and key directly, as in RoPE, we will introduce the notation $\alpha(\varphi(x_i, p_i), \varphi(x_j, p_j))$ for positional encoding function $\varphi$.

## 2.2 Absolute and Relative Positional Encoding

Absolute Positional Encoding (APE) is a common way of embedding token positions in transformers by adding position-dependent vectors, i.e. $\varphi(\mathbf{x}, p) := \mathbf{x} + \mathrm{PE}(p)$, where $\mathbf{x}$ is a token embedding, $p$ is its position, and $\mathrm{PE} : \mathbb{Z} \to \mathbb{R}^N$. Previous work has suggested learning a per-position token as PE [17, 20]. However, this restricts the network to fixed context length, removing the ability to extrapolate to different sequence lengths. The alternative is to add a deterministic function to the embedding. Vaswani et al. [73] proposed to add Fourier modes,

$$PE_n(p) = \begin{cases} \sin\left(p\,\omega_{\frac{n}{2}}\right), & \text{if } n \bmod 2 = 0 \\ \cos\left(p\,\omega_{\lfloor \frac{n}{2} \rfloor}\right), & \text{if } n \bmod 2 = 1, \end{cases} \tag{5}$$

where $n$ is a dimension within the positional embedding vector and $\omega_n$ is a frequency term which increases with dimension. Note that this pairs elements in the embedding vector with each pair being transformed by the same frequency.

For ease in future notation, we will use $D := N/2$ as the number of pairs and interpret the embedded token as a $D \times 2$ tensor. One can also interpret this tensor as representing the coefficients of a complex number, the first representing the real and second representing the complex part. Then we can succinctly write this form of positional encoding as

$$\varphi(\bar{\mathbf{x}}, p) = \bar{\mathbf{x}} + e^{i\boldsymbol{\omega} p}, \tag{6}$$

where we use $\bar{\cdot}$ to indicate complex-valued vectors, $\bar{\mathbf{x}} \in \mathbb{C}^D$. For this notation, we should also adjust the attention score for complex numbers,

$$\alpha(\bar{\mathbf{q}}, \bar{\mathbf{k}}) = \mathrm{Re}\left[\bar{\mathbf{q}}^\top \bar{\mathbf{k}}\right], \tag{7}$$

where $\bar{\mathbf{q}} = \bar{\mathbf{W}}_q \varphi(\bar{\mathbf{x}}, p)$ and $\bar{\mathbf{k}} = \bar{\mathbf{W}}_k \varphi(\bar{\mathbf{x}}, p)$, with $\bar{\mathbf{W}}_q, \bar{\mathbf{W}}_k \in \mathbb{C}^{D \times D}$, and $\top$ is assumed to be the Hermitian transpose. With Eq. 7 implied, we will continue with the notation in Eq. 3.

**Relative Positional Encodings** Positional embeddings rely on being able to assign position values to each token. However, how one assigns positions can often be arbitrary. One could just as correctly assign the first token the value zero and consider natural numbers, or assign the middle token of a corpus zero and consider integers. We can relax the assumption of a canonical way of labeling positions in APE by relying on relative distances between tokens, resulting in $\alpha_{ij} = \alpha(\mathbf{x}_i, \mathbf{x}_j, p_i - p_j)$. This is called relative positional encoding. We refer to this property as embeddings having a *relative positional bias*, or equivalently, having *shift-equivariance* (see Appendix E.1 for discussion of the equivalence). In this manuscript, we will simply use the term equivariance with the implication that the attention score is invariant to shifts in the query and key.

## 2.3 Rotary Positional Encodings (RoPE)

There are four common properties that are often preferred for positional embeddings: equivariance, key-query separability, linearity, and locality. For further details and why one may want these properties see Appendix E.

From the properties, Rotary Positional Embeddings (RoPE) were derived by Su et al. [69]. Rather than adding a positional embedding to the patch embedding, RoPE proposed to *modify the queries*

*and keys* by rotating them in pairs. By interpreting queries and keys as complex vectors, we can express this rotation as

$$\varphi(\bar{q}_d, p) = e^{i\omega_d p}\bar{q}_d \qquad\qquad \varphi(\bar{k}_d, p) = e^{i\omega_d p}\bar{k}_d. \qquad (8)$$

Since we assume the same operation is applied to the queries and keys, from now on, we will use $\mathbf{z}$ to refer to operations which act on both. In matrix form, this is given by $e^{i\omega_d p}$ can be represented as

$$e^{i\omega_d p} \equiv \begin{bmatrix} \cos(\omega_d p) & -\sin(\omega_d p) \\ \sin(\omega_d p) & \cos(\omega_d p) \end{bmatrix} = \mathbf{R}_{\omega_d p}, \qquad (9)$$

where $\mathbf{R}_{\omega_d p_t}$ is a rotation matrix. While the rotation matrix is more intuitive, the complex exponential form will be useful for the mathematics in Section 3, so we will alternate between the two. Recall, we use the convention that real valued queries and keys will have dimension $N$ and the complex interpretation will have dimension $D$.

One can represent the effect of RoPE as the application of a block diagonal of rotation matrices,

$$RoPE(\mathbf{z}, p) = \mathbf{R}_p \mathbf{z} = \begin{bmatrix} \mathbf{R}_{p\omega_1} & \mathbf{0} & \cdots \\ \mathbf{0} & \mathbf{R}_{p\omega_2} & \cdots \\ \vdots & \vdots & \ddots \end{bmatrix} \begin{pmatrix} \mathbf{z}_1 \\ \vdots \\ \mathbf{z}_D \end{pmatrix} \equiv \begin{bmatrix} e^{ip\omega_1} & 0 & \cdots \\ 0 & e^{ip\omega_2} & \cdots \\ \vdots & \vdots & \ddots \end{bmatrix} \bar{\mathbf{z}}, \qquad (10)$$

where $\mathbf{z}_d$ is a query pair. We introduce this block-diagonal form as it was the notation used in Su et al. [69]. However, we will primarily stick to the index notation in Eq. 8.

## 2.4  2D RoPE Embeddings

RoPE is constrained to operate on (1D) sequences. Motivated by the success of RoPE in language modeling, there have been growing efforts to extend it to multi-dimensional positions [10, 15], which we outline below. We will use $M$ to refer to the dimensionality of the position, but will primarily focus on images, where $M = 2$ and $p_i, p_j \in \mathbb{R}^2$.

**Trivial 2D RoPE.**   One could trivially encode $\mathbf{p} = (p_x, p_y)$ using rotation matrices $\mathbf{R}_{\omega_d p_x}, \mathbf{R}_{\omega_d p_y}$:

$$\varphi(\mathbf{z}_d, \mathbf{p}) = \mathbf{R}_{\omega_d p_x}\mathbf{R}_{\omega_d p_y}\mathbf{q}_d = \mathbf{R}_{\omega_d(p_x+p_y)}\mathbf{z}_d. \qquad (11)$$

However, in this case all positions with $p_x + p_y = c$ would get the same positional encoding $\mathbf{R}_{\omega_d c}$.

**Axial RoPE.**   More practically, RoPE is extended to multiple dimensions by letting $x$ and $y$ act on different dimensions,

$$\varphi(\mathbf{z}_d, \mathbf{p}) = \begin{bmatrix} \mathbf{R}_{p_x \omega_d} & \mathbf{0} \\ \mathbf{0} & \mathbf{R}_{p_y \omega_d} \end{bmatrix} \mathbf{z}_d, \qquad (12)$$

where queries and keys are now split into four-dimensional vectors, $\mathbf{z}_d^\top = \left[ z_1^{(x)}, z_2^{(x)}, z_1^{(y)}, z_2^{(y)} \right]$. The block-diagonal matrix can once again be viewed as a tensor of shape $N/2M \times M \times 2$, where $M$ is once again the dimensionality of position – in this case $M = 2$ for horizontal and vertical position. This gives the index notation

$$\varphi(\mathbf{z}_{d,m}, p_m) = \mathbf{R}_{\omega_{d,m} p_m}\mathbf{z}_{d,m}, \qquad (13)$$

for $m \in \{x, y\}$. From a programming perspective, one can interpret this as a form of batched matrix multiplication.

While this method eliminates the symmetry, it treats $x$ and $y$ as independent. The result is a separable attention score of the form

$$\alpha_{i,j} = \alpha_{ij}^{(x)} + \alpha_{ij}^{(y)}, \qquad (14)$$

where $\alpha_{ij}^{(x)}$ and $\alpha_{ij}^{(y)}$ are components of the attention score which depend only on $p_x$ and $p_y$, respectively. The frequencies are restricted to the axes, hence it is called Axial RoPE. This over-emphasizes horizontal and vertical relationships at the expense of oblique directions creating gridded patterns shown in Figure 1. To represent oblique patterns, the rotations would have to be performed along directions that contain both an $x$ and a $y$ component, i. e. frequencies that are not aligned on the axis in Figure1. These frequencies have been referred to as "mixed frequencies" [26].

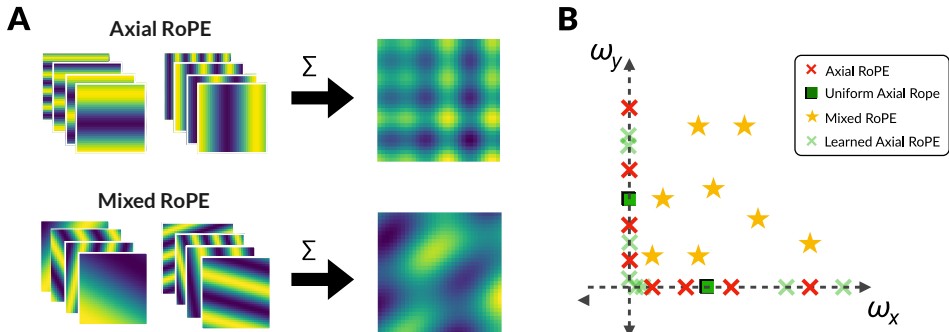

Figure 1: The attention patterns of Axial and Mixed RoPE. **A.** Each dimension pair in the query and key vectors is rotated based on the position creating an *attention pattern*. The pixel value of the attention pattern is $\alpha(\mathbf{x}_q, \mathbf{x}_k, \mathbf{p}, \mathbf{0})$, where $\mathbf{p} = (i, j)$ – the pixel location. On the left, the attention pattern of individual component-pairs in the embedding vector is shown and, on the right, the components are combined into the overall attention pattern for a randomly sampled query and key vector. **B.** Location of the rotations frequencies in 2D frequency space. Axial RoPE can only represent frequencies that lie on an axis resulting in the grid-like attention patterns. Unlike Axial RoPE, Mixed RoPE can assign different directions to each component-pair (**A** Bottom). When Axial RoPE uses fixed frequencies, the frequencies are spread exponentially. However, they can be implemented as learnable parameters. For Uniform RoPE, all frequencies are fixed to a single value for each axis.

**Mixed RoPE: Learned mixed frequencies.** The inclusion of mixed frequencies has empirically been shown to positively impact learning and generalization [26]. The naive approach in Eq. 11 is only a problem when $x$ and $y$ rotate by the same frequency in every dimension. One could instead parameterize the frequencies with two separate frequencies in each dimension,

$$\varphi(\mathbf{z}_d) = \mathbf{R}_{\omega_{dx}x}\mathbf{R}_{\omega_{dy}y}\mathbf{q}_d = \mathbf{R}_{\omega_{dx}x+\omega_{dy}y}\mathbf{q}_d. \tag{15}$$

By making the $\omega_x$ and $\omega_y$ parameters learnable, the attention pattern can learn mixed-frequency patterns by constructing a superposition of different diagonal patterns, as shown in Figure 1.

**LieRE.** Recently, RoPE has been interpreted through the lens of Lie algebras [44, 52, 60]. For an intuitive introduction to how Lie algebras appear, see Appendix E.3. Lie Rotary Position Encodings (LieRE) [52] extend Mixed RoPE by applying $N$-dimensional rotation matrices, rather than $2 \times 2$ matrices applied to pairs, using a linear combination of learned skew-symmetric Lie algebras,

$$\varphi(\mathbf{z}, p) = \exp(\mathcal{A}_x p_x + \mathcal{A}_y p_y)\mathbf{z}, \tag{16}$$

where $\exp$ is the matrix exponential, and the $\mathcal{A}$ terms are $N \times N$ skew-symmetric matrices – which are Lie group *generators* of a subgroup of $SO(N)$. Mathematically, LieRE is the most general rotary-based embedding method as skew-symmetric matrices are the generators of any $N$-D rotation. However, unlike the other two methods, LieRE is not guaranteed to be equivariant.

## 3   The Generality of Learned RoPE and Mixed RoPE

While LieRE is motivated as generalizing RoPE to $M$-D rotations, in this section we will show that LieRE in one dimension can be learned by implementing RoPE with parameterized frequencies. For $M$-D positions, LieRE is not equivariant unless the generators commute. If the generators are required to commute, we show that LieRE can be re-expressed as Mixed RoPE. Thus, we conclude Mixed RoPE to be a general solution for $M$-D equivariant rotary embeddings. In this section, we will give informal proofs focused on high-level insights.

### 3.1   1D-LieRE is equivalent to 1D RoPE with learned frequencies

In this section, we prove that any one-dimensional LieRE can be expressed as RoPE with parameterized rotation frequencies. Thus, we conclude RoPE to be a computationally efficient way of expressing a $D$-dimensional rotation, i.e., 1D-LieRE.

> **Proposition 1.** *Any 1D-LieRE can be parameterized by RoPE with learned frequencies.*

To see why Proposition 1 holds, suppose we have a 1D-LieRE embedding with a learned generator $\mathcal{A}$. By formulation, $\mathcal{A}$ is skew-symmetric, $\mathcal{A}^\top = -\mathcal{A}$. The positionally encoded attention between query $\mathbf{q} = \mathbf{W_q x}$ and $\mathbf{k} = \mathbf{W_k x}$ is

$$\alpha(x_i, x_j, p_i, p_j) = (\exp(\mathcal{A}p_q)\mathbf{q})^\top \exp(\mathcal{A}p_k)\mathbf{k}. \tag{17}$$

Any skew-symmetric matrix has an eigenvalue decomposition $\mathcal{A} = \mathbf{U}\mathbf{\Lambda}_{\mathcal{I}}\mathbf{U}^\top$ where $\mathbf{\Lambda}_{\mathcal{I}}$ is a diagonal matrix of purely imaginary (or zero) eigenvalues and $\mathbf{U}$ is a unitary matrix, $\mathbf{U}^\top\mathbf{U} = \mathbb{I}$. Moreover, the matrix-exponential of an eigenvalue decomposition simplifies to $\exp(\mathbf{U}\mathbf{\Lambda}_{\mathcal{I}}\mathbf{U}^\top) = \mathbf{U}\exp(\mathbf{\Lambda}_{\mathcal{I}})\mathbf{U}^\top$. This allows us to express attention as

$$\alpha(x_i, x_j, p_i, p_j) = \mathbf{q}^\top \mathbf{U} \exp(-p_q\mathbf{\Lambda}_{\mathcal{I}})\mathbf{U}^\top\mathbf{U}\exp(p_k\mathbf{\Lambda}_{\mathcal{I}})\mathbf{U}^\top\mathbf{k} \tag{18}$$

$$= \mathbf{q}'^\top \exp(-p_q\mathbf{\Lambda}_{\mathcal{I}})\exp(p_k\mathbf{\Lambda}_{\mathcal{I}})\mathbf{k}', \tag{19}$$

where $\mathbf{q}' = \mathbf{W}'_q\mathbf{x}$ with $\mathbf{W}'_q = \mathbf{U}\mathbf{W}_q$, and $\mathbf{k}' = \mathbf{W}'_k\mathbf{x}$ with $\mathbf{W}'_k = \mathbf{U}^\top\mathbf{W}_k$. Because the eigenvalue matrix is diagonal, the exponential is given by

$$\exp(p\mathbf{\Lambda}_{\mathcal{I}}) = \begin{bmatrix} e^{i\lambda_0 p} & 0 & \dots & 0 \\ 0 & e^{i\lambda_1 p} & \ddots & \vdots \\ \vdots & \ddots & \ddots & 0 \\ 0 & \dots & 0 & e^{i\lambda_{N-1} p} \end{bmatrix}. \tag{20}$$

Notice that this is the same as the complex formulation of RoPE defined in Eq. 10, where the eigenvalues of the generator correspond to the rotation frequencies of the rotation matrices. Thus, *any* 1D-LieRE can be expressed as RoPE with learnable frequencies by absorbing the matrix of eigenvectors of $\mathcal{A}$ into the weight matrices $\mathbf{W_q}$ and $\mathbf{W_k}$. Since 1D-LieRE learns a rotation in $SO(D)$, RoPE can be seen as an efficient way to represent a rotation in $\mathbb{R}^D$.

## 3.2 Extending RoPE to more than one dimension

While this proof works for 1D positions, it does not generalize to $M$-D without introducing extra inductive biases or giving up equivariance. By imposing constraints on $\mathcal{A}_x$ and $\mathcal{A}_y$, we can categorize the other RoPE methods based on the assumptions made.

**Generators rotate independent subspaces.** For example, one can impose the assumption that $p_x$ and $p_y$ rotate independent subspaces in $\mathbb{R}^N$. Mathematically, this assumption would imply that

$$\forall d \in [1, D] : \lambda_d^{(x)} = 0 \text{ or } \lambda_d^{(y)} = 0, \tag{21}$$

where $\lambda_d^{(x)}$ and $\lambda_d^{(y)}$ are the eigenvalues of $\mathcal{A}_x$ and $\mathcal{A}_y$, respectively. This is equivalent to rotating independent components of the query/key as done by Axial-RoPE.

**Commutative generators.** For LieRE to be equivariant, we only need to ensure that the generators commute. If we make this assumption, then we arrive at Mixed RoPE.

> **Proposition 2.** *Any $M$-dimensional LieRE with commutative generators can be parameterized by Mixed RoPE.*

To see why Proposition 2 holds, suppose we can diagonalize $\mathcal{A}_x = \mathbf{U}_x\mathbf{\Lambda}_x\mathbf{U}_x^\top$ and $\mathcal{A}_y = \mathbf{U}_y\mathbf{\Lambda}_y\mathbf{U}_y^\top$. If we take the assumption that $\mathcal{A}_x$ and $\mathcal{A}_y$ commute,

$$\mathcal{A}_x\mathcal{A}_y = \mathcal{A}_y\mathcal{A}_x \implies [\mathcal{A}_x, \mathcal{A}_y] = [\mathcal{A}_y, \mathcal{A}_x] = 0. \tag{22}$$

$[\mathcal{A}_x, \mathcal{A}_y] = \mathcal{A}_x\mathcal{A}_y - \mathcal{A}_y\mathcal{A}_x$ is the Lie bracket. This implies that $\mathcal{A}_x$ and $\mathcal{A}_y$ are *simultaneously diagonalizable* (Lemma 2). Thus, commutativity implies that $\mathbf{U}_x = \mathbf{U}_y := \mathbf{U}$. We can write

$$\mathbf{A} = \exp(\mathcal{A}_x p_x + \mathcal{A}_y p_y) = \mathbf{U}\exp(\mathbf{\Lambda}_x p_x + \mathbf{\Lambda}_y p_y)\mathbf{U}^\top, \tag{23}$$

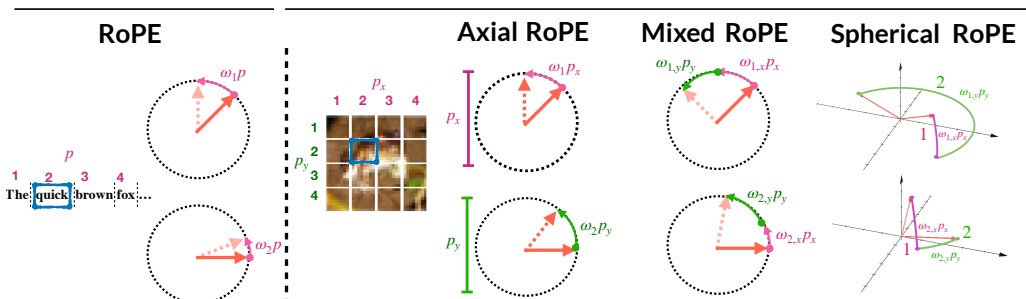

Figure 2: Diagram of each rotary embedding's effect on the subvector, $\mathbf{z}_d$. While Mixed RoPE affects 2D vector pairs, Spherical RoPE affects 3D vector triplets. Axial RoPE rotates independent dimensions for $p_x$, thus containing *pairs of pairs*, or effectively quadruples. Each $\mathbf{z}$ contains $D$ sub-vectors rotating at different frequencies. While the order in which the rotations are applied does not matter for Axial or Mixed RoPE, order matters for Spherical RoPE. Explicitly, the triplet is first rotated around the axis associated with $p_x$ and then rotated around the axis associated with $p_y$.

which leads to Eq. 15. Thus, Mixed RoPE forms a general solution for assumptions of commutativity, which is necessary for LieRE to be relative. This also mathematically shows Mixed RoPE to be the strict generalization of Axial RoPE.

In summary, learned frequency RoPE embeddings represent an efficient way of learning a much more general set of $SO(D)$ than is commonly believed. However, in order to generalize to higher dimensions while retaining their status as relative positional encodings, assumptions must be made. Mixed RoPE generalizes this to $M$-D positions, spanning the entire solution class for relative LieRE. Thus, LieRE-like methods with commutative $\mathcal{A}_x$ and $\mathcal{A}_y$ – such as in Schenck et al. [60] – are not more expressive than Mixed RoPE and any empirical differences in performance must be attributed to the learning dynamics due to different parameterizations. However, it remains unclear whether equivariance is the real reason for RoPE's success.

## 4 Experiments

When extending RoPE to more than one dimension, we must either constrain ourselves to commuting Lie algebras or give up relativity. We therefore ask the question: Why does RoPE work? Which properties should be preserved for generalizing RoPE to vision? To explore this question, we propose two new RoPE variants: *Spherical RoPE*, which takes a non-commutative assumption, and *Uniform-Frequency RoPE*, which uses a single fixed rotation frequency across all dimensions. Below we provide a high level outline the different embeddings. We compare the existing positional embedding methods APE [17], Axial RoPE [15], Mixed RoPE [10], and LieRE [52] to these two new variants to understand whether equivariance, oblique directions or a variety of spatial frequencies are important features of PEs for vision.

**Spherical RoPE.** We propose Spherical RoPE as a method between Mixed RoPE and LieRE that minimally changes 2D RoPE to break equivariance. Spherical RoPE embeds position as

$$\varphi(\mathbf{z}_d, \mathbf{p}) = \mathcal{Y}_{\omega_{dx}x}\mathcal{R}_{\omega_{dy}y}\mathbf{q}_d, \tag{24}$$

where $\mathbf{q}_d \in \mathbb{R}^3$ is now a triplet instead of a pair, and $\mathcal{Y}$ is a block diagonal of $3 \times 3$ *yaw* matrices and $\mathcal{R}$ is a block diagonal of *roll* matrices.

$$\mathcal{Y}_{\omega_{dx}x} = \begin{bmatrix} \cos(\omega_{dx}x) & -\sin(\omega_{dx}x) & 0 \\ \sin(\omega_{dx}x) & \cos(\omega_{dx}x) & 0 \\ 0 & 0 & 1 \end{bmatrix} \qquad \mathcal{R}_{\omega_{dy}y} = \begin{bmatrix} 1 & 0 & 0 \\ 0 & \cos(\omega_{dy}y) & -\sin(\omega_{dy}y) \\ 0 & \sin(\omega_{dy}y) & \cos(\omega_{dy}y) \end{bmatrix}. \tag{25}$$

Intuitively, rather than RoPE rotating around a circle, Spherical RoPE rotates around a sphere using Euler angles.

Importantly, spherical rotations like LieRE are *non-commutative* making them *not equivariant*. In fact, their generators are strictly *non-commutative*, $\mathcal{A}_x\mathcal{A}_y \neq \mathcal{A}_y\mathcal{A}_x$. While non-commutativity does not mean Spherical RoPE is incapable of learning or approximating equivariance throughout the network, it is the component of LieRE removed by Mixed RoPE and works which enforce commutativity such as Yu et al. [85] and Schenck et al. [60].

Table 1: Table listing the properties of each of the rotary-based methods.

| Positional Encoding | Vision | Strictly Equivariant | Oblique Directions | Requires Learning |
|---|---|---|---|---|
| Rotary (RoPE) [69] | ✗ | ✓ | N/A | ✗ |
| Axial RoPE [68] | ✓ | ✓ | ✗ | ✗ |
| Mixed RoPE [26] | ✓ | ✓ | ✓ | ✓ |
| LieRE [52] | ✓ | ✗ | ✓ | ✓ |
| Spherical RoPE | ✓ | ✗ | ✓ | ✗ |
| Uniform RoPE | ✓ | ✓ | ✗ | ✗ |

We hypothesized Spherical RoPE to have a number of advantages. While Axial RoPE is unable to express oblique directions, Spherical RoPE can. Like Axial RoPE, Spherical RoPE can use fixed frequencies making it computationally cheaper than LieRE and Mixed RoPE since sines and cosines of the frequencies can be precomputed. However, our main interest is that Spherical RoPE is comparable in terms of expressivity to Mixed and Axial RoPE while being non-equivariant.

**Uniform-Frequency RoPE.** For an initial evaluation on the impact of relative position, we propose Uniform-Frequency RoPE. For this method, we perform Axial RoPE with a single frequency shared across all rotation matrices. While still being relative, this serves as a more restricted version of RoPE. If this method performs significantly worse than other methods, it indicates the importance of having a range of frequencies. We implement uniform frequencies for Axial RoPE to gauge against relative importance of equivariance.

In one extreme, the rotation frequency could be zero resulting in no changes to the queries and keys. In the other extreme, the frequency could be set very high resulting in large changes to the queries and keys. As a note, it is the frequency relative to the resolution of the image that is important. Frequencies higher than the sampling rate are equivalent to low frequencies. To ensure every position has a unique encoding, we fix the frequency to perform one rotation cycle across the entire image.

**Datasets and architecture.** We test the different PEs on CIFAR100 [39] and ImageNet [58] using a standard Vision Transformer – the ViT-S implementation from the timm [79] library. For Learned APE, we use the baseline ViT-S which uses learned positional encodings rather than sinusoidal. We follow much of the DeiT-III training procedure proposed in Touvron et al. [72]. However, for ImageNet, we do not use dropout, MixUp, or CutMix as we observed that they significantly increase the number of epochs necessary for convergence. For ImageNet, we evaluate models trained after 200 epochs and 400 epochs for CIFAR100. We evaluate without any hyperparameter tuning directly on the validation sets. For further details on hyperparameters and experimental setup, see Appendix H. Error bars were created using three models with different random seeds.

**Generalization to larger image sizes.** We also perform an experiment to test how well different PEs generalize across image sizes. Our approach to this experiment follows prior research [26, 52]. The learned embeddings in Learned APE cannot be extrapolated, so we interpolate new embeddings when changing the number of patches. For RoPE embeddings, we take square dimensions and parameterize position such that the top-left corner of the image corresponds to $p_x = p_y = -\pi$ and the bottom-right corner correspond to $p_x = p_y = \pi$ with all other positions are evenly spread between the two for training. When increasing the image size, we extrapolate by scaling the range by the ratio of the new image size to the training image size while keeping the patch size constant.

**Additional Evaluations** Additional evaluations can be found in Appendix I including method speeds, experiments with smaller data splits, a segmentation task, and evaluation of the learned weights.

Table 2: Performance comparison (top-1 accuracy) across datasets and methods.

| | Top-1 Accuracy (%) | |
|---|---|---|
| **Fixed Encoding** | **CIFAR100** | **ImageNet** |
| Learned APE | $64.2_{\pm 0.9}$ | $72.7_{\pm 0.1}$ |
| Axial RoPE | $72.1_{\pm 0.6}$ | $75.6_{\pm 0.2}$ |
| Uniform RoPE (Our Ablation) | $70.5_{\pm 0.2}$ | $74.9_{\pm 0.3}$ |
| Spherical RoPE (Our Ablation) | $\mathbf{73.2}_{\pm 0.4}$ | $\mathbf{76.4}_{\pm 0.3}$ |
| **Learned Encoding** | | |
| Learned Axial RoPE | $72.9_{\pm 0.6}$ | $75.7_{\pm 0.4}$ |
| LieRE | $73.1_{\pm 0.2}$ | |
| Mixed RoPE | $\mathbf{74.7}_{\pm 0.3}$ | $\mathbf{77.4}_{\pm 0.1}$ |
| Learned Spherical RoPE (Our Ablation) | $74.1_{\pm 0.4}$ | $\mathbf{77.4}_{\pm 0.2}$ |

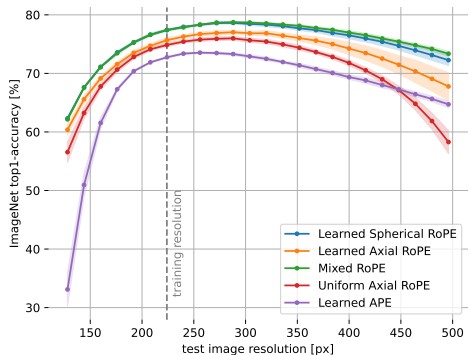

Figure 3: Dependence of accuracy on image resolution for ViT-S with various positional embedding methods on ImageNet-1k. Error bars reflect the standard deviation across three models trained with different seeds.

# 5 Results

To evaluate the importance of different properties of positional embeddings in vision transformers, we trained the same ViT with different positional embeddings on CIFAR100 and ImageNet-1K. We start by evaluating the models on images of the same resolution as during training. If equivariance is important, we would see Axial and Mixed RoPE to perform better than Spherical RoPE, which lacks strict equivariance. On the other hand, if oblique frequencies are important, then we would observe Mixed and Spherical RoPE to do better than Axial RoPE, which does not capture oblique directions. We find that the lack of equivariance does not hinder Spherical RoPE. It outperforms Axial RoPE and performs comparably to Mixed RoPE. Moreover, we would expect equivariant methods to be especially effective in the low data regime. However, in Appendix I we observe Learned Spherical RoPE performs the best despite its lack of inductive bias. This suggests that the benefits to performance and generalization on ImageNet for Mixed RoPE may be due to its extra parameters. However, Axial RoPE and Uniform RoPE perform significantly worse suggesting oblique directions to be more important than equivariance.

When comparing with absolute positional encodings, we observe that all forms of RoPE perform better than learned APE (Table 2). This includes Uniform RoPE, the variant that uses only a single frequency. Moreover, all forms of RoPE using diverse frequencies outperform Uniform RoPE and have similar performance (whether they are learned or not), suggesting that diversity of frequencies is important. Spherical RoPE adheres much closer to the vectorized implementation of other RoPE methods than LieRE. As our goal was primarily to identify the most impactful properties of $M$-D RoPE and not maximize accuracy, none of our conclusions depend on precise performance numbers for LieRE.

Last, we asked how well different PEs generalize across image sizes. Equivariance is often thought to aid model generalization. However, when evaluating each model using higher resolutions images, i. e. increasing the number of patches, we found Spherical RoPE to be the most effective method (Figure 3), suggesting equivariance may not be the reason for RoPE's generalization.

# 6 Discussion

Because we see very little variation between Spherical RoPE and Mixed RoPE, we conclude that equivariance is only a minor contributor to the increased performance seen by RoPE for vision. In fact, Spherical RoPE appeared to extrapolate to higher resolutions better than Axial RoPE. This could suggest that oblique frequencies are important for extrapolation. However, extrapolation is only done on short length scales, so this may not hold in language.

There are two important differences between vision and language transformers: context length and patch variation. Where LLMs have on the order of 128K context windows [22], vision transformers

only have $16 \times 16$ patches. Moreover, patches have more variation as tokens than language tokens, thus allowing the content embeddings to store information about the relevance of oblique directions. Because the context size is small, we hypothesize that there could be methods that perform better than Mixed RoPE and are more general than LieRE for vision. While LieRE was proposed with skew-symmetric generators to generalize RoPE to $N$-D rotations, Lie algebras do not have to be skew-symmetric. The skew-symmetry is important for maintaining numerical stability over long contexts [69]. However, skew-symmetry also results in Proposition 3.1 of Barbero et al. [5], which proves RoPE to be non-local. Since the context size is small for images, numerical stability is likely not an issue, thus freeing the space of Lie algebras available to us – including Lie algebras that encourage locality.

We observe a decrease in generalization when using uniform frequencies. This finding qualifies Barbero et al. [5]'s hypothesis that the various semantic lengths contribute to RoPE's performance. However, Uniform RoPE outperformed learned APE, suggesting the reason why RoPE performs well is not among the properties we tested. We speculate this could be a flaw in additive positional embeddings themselves; additive methods create a trade-off between the magnitude of position and content – forcing tokens that vary significantly with position to have lower magnitude to be closer to the origin.

# 7  Conclusion

We conclude that Mixed RoPE is a very general solution for $M$-D data if equivariance is a necessity. However, we see little evidence that strict relative positional bias is impactful for vision transformers. However, RoPE methods have still been found to greatly improve performance in ViTs. Thus, we conclude that evidence suggests that RoPE does not *need* strict equivariance constraints to boost performance over APE methods.

# 8  Acknowledgments

This project has received funding from the European Research Council (ERC) under the European Union's Horizon Europe research and innovation programme (Grant agreement No. 101041669). We gratefully acknowledge the computing time granted by the Resource Allocation Board and provided on the supercomputer Emmy/Grete at NHR-Nord@Göttingen as part of the NHR infrastructure (project nim00012).

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

# A    Broader Impact

This work is fundamental research. While this work could lead to the discovery of better positional encodings and higher performing visual foundation models, the positivity or negativity of this impact is determined by the downstream task and not this work.

# B    Limitations

While our results do not show relative embeddings to be detrimental, we believe them to be evidence that equivariance is not the reason for RoPE's success. However, our experiments were performed in Vision where the number of tokens is limited compared to the long context lengths of NLP. Moreover, the datasets are not what many believe to be "at scale". While Spherical RoPE and LieRE would intuitively be favored at scale over Axial RoPE, as they have less inductive bias, it is unclear whether inductive bias and equivariance is favored at scale [9].

It has also been shown that vision is *not* a purely equivariant task and benefits from relaxed equivariance [18]. Our results do not show that equivariance is not useful in tasks that are grounded in physics and obey strict symmetries.

# C    Literature Review

## C.1    Natural Language Processing

In natural language, positional encoding has been used to break the permutation, "bag of words", symmetry [73]. Although this could be done by learning a vector per position, this is both memory-expensive for large context sizes making it practical to apply to only the first layer. Moreover, it does not allow for extrapolation at test time to context sizes beyond training. Thus, it is favorable to perform positional embeddings with a predictable deterministic function. One way of doing this is to make the attention relative with local receptive fields, as is done implicitly in convolutional neural networks [12]. Sinusoidal positional embeddings were proposed due to approximate local and shift-invariant properties of Random Fourier Features [57]. Since sinusoidal, other methods have been proposed to get guaranteed shift invariance by explicitly parameterizing based on distance [64, 54, 55]. However, these methods require a positional embedding for every pair of positions which is not supported by many of the efficient attention optimizations such as Flash attention [16] [3].

Rotary Positional Embeddings (RoPE) have become the staple in NLP having recently been adopted by many of the large language models [76, 22, 71, 43, 28]. However, these methods also use causal masking, which has been shown to allow models with no positional embedding to recover absolute position [24, 82, 75, 33]. This has led to questions on the importance of relative position [5].

In language, there has also been extensions to RoPE proposed through NTKs and kernel methods [11]. However, these methods have not, to our knowledge, seen use in vision.

## C.2    Vision and Video

Vision transformers were introduced in Dosovitskiy et al. [17] and, though they tried sinusoidal position encodings, found learnable position encodings to perform best. For convolution-esque models such as SWin transformers, relative positional encodings have been popular [47, 14]. More recently, RoPE has been shown to be an efficient and simple way to have relative embeddings and has been extended to 2D using Axial and Mixed RoPE. Going beyond 2D to Video data, Axial RoPE has become increasingly popular. The extension was first attributed to Wang et al. [76] as 3D-RoPE or M-RoPE, leading to two separate Video-RoPE papers from Wei et al. [77] and Liu et al. [48]. Both of these focus on the order of the position enumeration and interleaving positions. However, this should not be a problem if frequencies are not deterministic, *or* if frequencies are indexed by both $d$ and modality $m$ as done in Eq 13. We highly recommend using either Mixed RoPE or LieRE which extend naturally for videos.

LieRE embeddings have thus far been the most general form of RoPE to $N$-D. However, Schenck et al. [60] has claimed the method to have a large memory footprint and proposed STRING. This

paper, a preprint released concurrently with the writing of this manuscript, follows much of the same math as this paper. However, they did not recognize that an orthogonal matrix is implicitly learned by the query and key matrix. Moreover, their method relies on commuting Lie algebras. From our insights in Section 3, their method can likely be viewed as a slower implementation of $N$-D Mixed-RoPE.

It is also worth noting that positional encodings have also been explored within vision through the area of Neural Fields [80]. Traditional coordinate MLPs have been found to be biased toward low-frequency functions [70] leading to more advanced positional encodings such as Random Fourier Features [57] or sinusoidal activation functions [65]. These implicit functions have been used to encode attention and message passing in graph neural networks with recent work being put in to make these functions equivariant to symmetry transformations [59, 8, 36].

### C.3    Graphs and AI in Science

Positional encodings are well studied within graph neural networks [42, 53]. Graphs are limited in their expressivity up to the Weisfeiler-Lehman (WL) graph isomorphism test [81], so positional encodings can break the isomorphism symmetry [25, 83]. Within this community, they propose *spectral attention* and graph Laplacians for positional encoding [38]. These methods seem extremely close to our analysis of RoPE, but from a very different perspective. We show that the frequencies of RoPE can be interpreted as the eigenvalues of an orthogonal transformation by taking the spectral decomposition.

In an overlapping vein, relative position encodings have been studied in terms of equivariant graph neural networks, often for scientific disciplines such as molecular physics [8, 62] or drug discovery [29]. One method to achieve equivariance is through defining relative coordinate frames [37]. This corresponds to the learned relative positional method described in Shaw et al. [64], but can be generalized to higher dimensions and different transformation using bi-invariant distance functions [6, 36, 78]. The message-passing functions of these works correspond to a generalization of attention scores [21].

However, even in these tasks with physics-grounded symmetries, the need for equivariance is hotly debated. While AlphaFold [29] was originally touted as the example of the success of equivariant inductive biases in science, AlphaFold 3 [1] explicitly stated that they benefited from removing this inductive bias at scale. However, while the harm of inductive bias at scale is the prevalent zeitgeist, it is not an established fact [9].

### C.4    Computational Neuroscience

Coupled oscillators have become a growing area of interest within computational neuroscience [34, 35, 67]. By observing the projection of the RoPE circles onto the real axis, one can interpret RoPE as time progression in $D$ uncoupled, undamped harmonic oscillators. This perspective naturally connects RoPE to Löwe et al. [49]'s series of papers on complex autoencoders and their extensions [50, 51].

In another, vein of research, there has been some work in hyper-dimensional computing[30, 31] in Phasor and Residue VSAs [40] which represent concepts as rotations around unit circles in high-dimensional spaces. These representations have strong connections with RoPE. Additionally, progress has been made in hypothesizing how biological neural networks encode positional knowledge with hexagonal grid cells, which can be represented as a discrete sum of three periodic functions oriented at the cubic roots of unity[66].

### C.5    Generality of RoPE

The generality of RoPE has been found by others. Schenck et al. [60], Su [68], and Liu and Zhou [44] all propose proofs similar to Proposition 1. However, Schenck et al. [60] miss that the orthogonal transformation can be incorporated into key matrix. Liu and Zhou [44] and Su [68] take the assumption of *reversibility*, which leads to the independent eigenvalue assumptions of Axial RoPE. All three works take the assumption of an abelian subgroup – i. e. commutative generators, – but miss the generality of Mixed RoPE. While Su [68] propose quaternions – i. e. spherical rotations – as a direction, they immediately dismiss it as a *no-go* because they lack equivariance. This exemplifies the

"circular argument," where equivariance is assumed to be necessary because work will not investigate non-equivariant positional encodings because equivariance is necessary.

Because our derivation was found independently of these works and the previous works are, to our knowledge, not published, we have left in Proposition 1. We would like to acknowledge their work, but retain the flow of this paper.

## D  Notation

| Symbol / Term | Dimension | Meaning | Notes |
|---|---|---|---|
| $\mathbf{x}_i$ | $\mathbb{R}^D$ | Patch/token/content vector of token $i$ | Raw input embedding |
| $x_i$ | $\mathcal{X}$ | Abstract content of token $i$ | Raw input embedding |
| $p_i$ | $\mathbb{R}^M$ or $\mathcal{P}$ | Position of token $i$, can be $M$-D or abstract $\mathcal{P}$ | Scalar (1D) or vector (2D) |
| $m$ | $\mathbb{Z}$ | Modality index | e.g., $x$, $y$, time |
| $M$ | $\mathbb{Z}$ | Number, or space, of Modalities | |
| $D$ | $\mathbb{Z}$ | Hidden dimension | Number of pairs/triples/quadruples |
| $T$ | $\mathbb{Z}$ | Number of Tokens | |
| $\mathbf{W}_q, \mathbf{W}_k, \mathbf{W}_v$ | $\mathbb{R}^{\mathcal{X} \times D}$ | Query, Key, Value Matrices | |
| $\mathbf{q}$ | $\mathbb{R}^N$ | $\mathbf{q}_i = \mathbf{W}_q x_i$ | Query vector |
| $\mathbf{k}$ | $\mathbb{R}^N$ | $\mathbf{k}_j = \mathbf{W}_k x_j$ | Key vector |
| $\mathbf{v}$ | $\mathbb{R}^N$ | $\mathbf{v}_j = \mathbf{W}_v x_j$ | Value vector |
| $\mathbf{Q}, \mathbf{K}, \mathbf{V}$ | $\mathbb{R}^{T \times N}$ | Query, Key, Values | $T$ tokens, $D$ latent dimensions |
| $\varphi(x, p)$ | $\mathcal{X} \times \mathcal{P} \to \mathbb{R}^D$ | Positional Encoding function | |
| $\mathbf{Z}$ | $\mathbb{R}^{T \times N}$ | Output of Attention | $\mathbf{Z} = \text{Attention}(\mathbf{Q}, \mathbf{K}, \mathbf{V})$ |
| $a(i, j)$ | $\mathbb{R}$ | Attention weight | Softmax of attention scores |
| $\alpha(\mathbf{q}, \mathbf{k})$ | $\mathbb{R}$ | Attention score | Inner product $\mathbf{q}^\top \mathbf{k}$ |
| $\omega_d/\lambda_d$ | $\mathbb{R}$ | Rotation frequency for dimension $d$ | Equivalent to eigenvalue of generator |
| $\mathbf{q}_d$ | $\mathbb{R}^{2/3/4}$ | Query pair/triple/quadruple at dimension $d$ | After RoPE or LieRE applied |
| $\mathbf{R}_{\omega_d p}$ | $\mathbb{R}^{2 \times 2}$ | $2 \times 2$ rotation matrix | Rotation based on frequency and position |

Table 3: Summary of Notations and Key Concepts

| Positional Encoding | Vision | Learned | Extrapolation | QK Separable | Relative | Linear Flow | Used In |
|---|---|---|---|---|---|---|---|
| Absolute (Sinusoidal) | ✗ | ✓/✗ | ✓ | ✓ | ✓ | ✗ | Transformer[73] |
| Absolute (Learned) | ✓ | ✓ | ✗ | ✓ | ✓ | ✗ | BERT, GPT, ViT[17] |
| Absolute (Random-Fourier) | ✗ | ✗ | ✓ | ✓ | ✗ | ✓ | FNet[41], Performer [13] |
| Relative (Learned) | ✗ | ✓ | ✗ | ✗ | ✗ | ✗ | Transformer-XL, T5 [56] |
| ALiBi | ✗ | ✓/✗ | ✓ | ✓ | ✓ | ✓ | LLaMA 2 [22], ALiBi [55] |
| NoPE | ✗* | ✗ | ✓* | ✓* | ✓* | ✓* | LLaMA 4 [2] |
| Rotary (RoPE) | ✗ | ✗ | ✓ | ✓ | ✓ | ✓ | Contemporary LLMs [76, 22, 71, 28] |
| Axial RoPE | ✓ | ✓/✗ | ✓ | ✓ | ✓ | ✓ | VisionLLaMA[15], Qwen2[76], VideoRoPE[77] |
| Mixed RoPE | ✓ | ✓ | ✓ | ✓ | ✓ | ✓ | Heo et al. [26] |
| LieRE | ✓ | ✓ | ✓ | ✓ | ✗ | ✓ | [52] |
| Spherical RoPE | ✓ | ✓/✗ | ✓ | ✓ | ✗ | ✓ | Ours |
| Uniform RoPE | ✓ | ✓/✗ | ✓ | ✓ | ✓ | ✓ | Ours |

Table 4: Comparison of positional encoding methods in transformer models. *NoPE makes some properties trivially true.

# E   Positional Encoding Properties

Rotary positional embeddings were derived in Su et al. [69] by drawing equations from assumed properties. While these appear as arithmetic assumptions and equations in their work, we formalize what properties these assumptions imply and why we may choose these assumptions in this section. In their paper, to derive their equations, they use equivariance (relativity), query-key separability of the positional encoding, linearity and incompressability, locality, and query-key symmetry.

1. Equivariance/Relativity: Attention score should be affected only by the relative position of two tokens, i. e. have the form

$$\alpha(x_i, x_j, p_i, p_j) = \hat{\alpha}(x_i, x_j, p_i - p_j). \tag{26}$$

2. Key-query seperability: The positional encoding, $\varphi$, of the query should not depend on the position of the key

$$\alpha(x_i, x_j, p_i, p_j) = \bar{\alpha}(\varphi(x_i, p_i), \varphi(x_j, p_j)) \tag{27}$$

3. Linearity: The positional encoding should be a linear flow, see Appendix E.3. Namely,

$$\varphi(\varphi(x, p_i), p_j) = \varphi(x, p_i + p_j). \tag{28}$$

4. Locality: The attention score between two tokens should decay with distance

$$\lim_{|p_i - p_j| \to \infty} \alpha(x_i, x_j, p_i, p_j) = 0 \tag{29}$$

## E.1   Relativity and Equivariant

We use the term *equivariant* interchangably with *relative*. Strictly speaking, one should specify the transformation or group you would like to be relative to, e. g. shift/rotation or $SO(2)$. As previous literature always refers to relative positional bias in terms of shifts/translations, in the main text, this is what we mean. We use the term equivariance to be the generalization of relativity beyond language because we would like to refrain from using the term "relativity" to describe the property of being a relative PE too often due to its connotation within theoretical physics. First, we define relative in the case of positional encodings in language as

$$\alpha(x_i, x_j, p_i, p_j) = \hat{\alpha}(x_i, x_j, p_i - p_j). \tag{30}$$

In the rest of this section, we mathematically explore where this equation comes from.

The behavior we are trying to capture is that if we renumber the words in the sentence, it should not affect the attentions score. Intuitively, if a text is padded with spaces at the beginning, that will not have a significant effect on the meaning of the sentences. We can ensure this by colloquially saying that the attention between two words should depend on the distance between them. Notice, that strictly speaking this is not a proper distance, since it can be negative; it is, instead, a *signed* distance function. Though this may seem pedantic in one dimension, in two dimensions defining a distance function is less unique. For example, one may choose $\mathbb{L}_1$ or $\mathbb{L}_2$ distance metrics. Because distance functions are more nebulous, it makes more sense to define relative in terms of the transformations that we would like our attention score to be independent of.

$$\alpha(x_i, x_j, p_i, p_j) = \alpha(x_i, x_j, T(p_i), T(p_j)). \tag{31}$$

These transformations can be combined to generate a set of transformations which leave the attention score unchanged, or *symmetric*. This set has the mathematical properties of a group and is known as a symmetry group. We can index transformations by elements in the symmetry group, $g \in G$, and let the elements act on

$$\alpha(x_i, x_j, p_i, p_j) = \alpha(x_i, x_j, g.p_i, g.p_j). \tag{32}$$

As an example, $g$ could represent an angle, $\theta$, and it may act on a vector $\mathbf{p}$ as a rotation $g.\mathbf{p} = \mathbf{R}_\theta \mathbf{p}$.

Connecting everything back to Eq. 30, Noether's theorem states that any continuous symmetry can be expressed as a conservation law. This allows us to introduce bi-invariant function [36, 78], or "Noether charge", $\beta(p_i, p_j)$, that is invariant under the group action,

$$\beta(p_i, p_j) = \beta(g.p_i, g.p_j) \implies \beta(p_i, p_j) - \beta(g.p_i, g.p_j) = 0. \tag{33}$$

Thus, we can express our symmetry group through isodistances of $\beta$,

$$\alpha(x_i, x_j p_i, p_j) := \hat{\alpha}(x_i, x_j, \beta(p_i, p_j)). \tag{34}$$

For example, we can pick the function

$$\beta(p_i, p_j) = p_i - p_j = (p_i - p_0) - (p_j - p_0) = \beta(p_i - o, p_j - p_0) \tag{35}$$

If we were to define $\beta(p_i, p_j) = |p_i - p_j|$, then we we would additionally be equivariant to reflection of the order of tokens in a sentence. If we trivially define $\beta(p_i, p_j) = C$, then we arrive at bag of words, or no positional encoding (NoPE). For a list of common transformations and their corresponding bi-invariants see Theorem 1 of Bekkers et al. [6].

## E.2 Query-Key Separability

Query and key separability is important for efficiency reasons. If we can decompose our positional encoded attention score as,

$$\alpha(x_i, x_j, p_i, p_j) = \alpha(\varphi(x_i, p_i), \varphi(x_j, p_j)) \tag{36}$$

then we can pre-compute the positional encoding for the queries and keys on time making the computation $O(T)$. If the positional encoding is not separable, then it will need to be computed for *every pair*, $(i, j)$[47, 56, 64]. Although there are many symmetries that can be exploited to make this not a quadratic computation, it removes the symmetries exploited by efficient attention mechanisms [7, 13, 32].

## E.3 Linear Flow Property

The property of being a "flow" was first proposed in Liu et al. [45], however it is not often discussed. It is a property inherently present in RoPE[69], LieRE[52] and ALiBi [55] embeddings, specifically as a *linear flow*.

We use the term *linear flow* for this property because the embedding can be found by repeated application of a linear function. However, the term "linear" this is a small misnomer because it is only *locally* linear. We define a *flow* as function

$$\varphi : \mathbb{R}^N \times \mathbb{R} \to \mathbb{R}^N \tag{37}$$

such that for all $x \in X$ and $p_1, p_2 \in \mathbb{R}$, the following conditions hold:

1. Initial condition (identity at time zero):

$$\varphi(0, x) = x \tag{38}$$

2. Group property (flow property):

$$\varphi(\varphi(\mathbf{x}, p_1), p_2)) = \varphi(x, p_1 + p_2) \tag{39}$$

3. Continuity (or differentiability): $\varphi$ is continuous with respect to its variables, depending on the context

Strictly speaking, continuity is not necessary for positional encodings as positions tend to be integer values. What we really wish to capture with this property is for the positional encoding to be recursively defined. It may be strange to wish to apply the positional encoding multiple times; however, by having the positional encoding as an endomorphism it can allow for more predictable behavior when extrapolating to larger contexts, which we suspect helps the model train.

We define a position embedding to be a *linear flow* if the flow has the form:

$$\varphi(\mathbf{x}, \Delta p) = \mathbf{A}\mathbf{x}, \tag{40}$$

for $\mathbf{A} \in \mathbb{R}^{N \times N}$ and $\mathbf{x} \in \mathbb{R}^N$, where $\Delta p$ is the increment rate for position. By Eq. 39, any position $p := p_0 \Delta p$ can then be attained by,

$$\varphi(\mathbf{x}, p) = \mathbf{A}^{p_0}\mathbf{x}. \tag{41}$$

This can be seen as a *geometric series* if $\mathbf{A}$ is a scalar as seen in Press et al. [55]. If we let $\Delta t$ become infinitesimal, then we can express the recurrence relationship as the ODE,

$$\frac{\partial \varphi}{\partial t} = \mathcal{A}\varphi \tag{42}$$

which we can integrate to get,

$$\varphi(\mathbf{x}, p) = \exp(\mathcal{A}p)\mathbf{x} \tag{43}$$

This $\mathcal{A}$ is our *generator* of the flow, which is also a generator for a *matrix Lie algebra*, which we focus on in the main text. The matrix exponential, $\exp : \mathbb{R}^{N \times N} \to \mathbb{R}^{N \times N}$, can be unstable for long contexts; similar to the scalar exponential function $e^{xp}$, the function can quickly become large for high values of $x$. However, this can be stable value $x = 0$, since it always results in one. Similarly, the matrix exponential can be stable if the divergence of the flow – trace of the generator – is zero. We call flow "incompressible" or "divergence-free" if the trace of $\mathcal{A}$ is zero, making the determinant of $\mathbf{A}$ unit. If fluid dynamics, this is called *incompressibility*. For fluids, this implies that the flow conserves mass.

If there are more than one generator of the Lie group, $\mathcal{A}_1$ and $\mathcal{A}_2$, then Eq. 39 must be modified to,

$$\varphi(\varphi(\mathbf{x}, \mathbf{p_1}), \mathbf{p_2})) = \varphi(\mathbf{x}, \mathbf{p_1} \circ \mathbf{p_2}), \tag{44}$$

where $\circ$ is the group product. By the Baker–Campbell–Hausdorff formula, $\exp \mathcal{A}_1 p_1 \exp \mathcal{A}_2 p_2 = \exp \mathcal{A}_1 p_1 + \mathcal{A}_2 p_2$ iff the commutator of $\mathcal{A}_1 p_1$ and $\mathcal{A}_2 p_2$ is zero, i. e. the matrices commute. If they do commute, then

$$\varphi(\varphi(\mathbf{x}, \mathbf{p_1}), \mathbf{p_2})) = \varphi(\varphi(\mathbf{x}, \mathbf{p_2}), \mathbf{p_1})) \implies \varphi(\mathbf{x}, \mathbf{p_1} \circ \mathbf{p_2}) = \varphi(\mathbf{x}, \mathbf{p_2} \circ \mathbf{p_1}) \tag{45}$$

thus making $\circ$ commutative and having the same properties as addition, $\circ := \text{“+"}$, and Eq. 39 will hold. In this case, the group/flow is known as an *abelian* Lie group, or *abelian flow*. However, if they do not commute, then $\circ$ will not commute and they are known as *non-abelian*. This also makes the flow *non-integrable*.

### E.4   Locality

Locality is often conflated with relativity. The general idea is that tokens far from each other should be independent of one another – i. e. attention should decay as distance grows. This often motivates the definition

$$\lim_{|p_i - p_j| \to \infty} \alpha(x_i, x_j, p_i, p_j) = 0 \tag{46}$$

for $p_i, p_j \in \mathbb{R}$ and $x_i, x_j \in \mathbb{R}^D$. However, this definition is *both* relative and local. We instead define local as,

$$\lim_{|p_i - p_0| \to \infty} \alpha(x_i, x_j, p_i, p_0) = 0. \tag{47}$$

The difference being that $p_0$ is the *origin* position. If an embedding is relative, then the origin is arbitrary and can be defined as $p_i$ or $p_j$. In Press et al. [55], they define the origin vector as the next word. However, they can only do this because of the causal mask.

In general, the most natural way to measure locality is through the concept of the quantum mechanical concept of the *variance of an operator*. We will simply use exponential decay, but we point interested readers to Chapter 3 of Griffiths [23]. This formalism works for RoPE as it is a linear transformation and the attention mechanism defines a Hilbert space.

To be clear, RoPE and LieRE are *not* local embeddings. This was shown for RoPE in Barbero et al. [5]. Because they are orthogonal matrices, they have unit determinant, which naturally precludes locality.

### E.5 Other properties

For completeness, there are two additional assumptions that are common.

**Adjoint symmetry of the Positional Encoding**  We implicitly assume that the positional encoding is symmetric for the query and key. That is, we assume that the query and key are from the same domain, so the positional encoding has the same representation. More generally, the positional encoding can act differently on the query and key,

$$\alpha(\bar{\varphi}(x_i, p_i), \varphi(x_j, p_j)) = \alpha(\varphi(x_i, p_i), \varphi(x_j, p_j)), \tag{48}$$

where $\bar{\varphi}$ is the positional encoding function for queries. More generally, we can have a relative embedding by letting $\bar{\varphi}$ act on queries differently from the keys. For example, if we let

$$\varphi(x, p) = \exp(\Lambda p) \qquad\qquad \bar{\varphi}(x, p) = \exp(-\Lambda p), \tag{49}$$

where $\Lambda$ is a diagonal matrix. We end up with,

$$\alpha(\bar{\varphi}(x_i, p_i), \varphi(x_j, p_j)) = \mathbf{q}_i^\top \exp(\Lambda(p_j - p_i))\mathbf{k}_j, \tag{50}$$

where RoPE can be interpreted as a simple harmonic oscillator, by weakening the symmetry requirement, one could incorporate damping. This can also be used to incorporate graph Laplacian positional encodings into the framework.

**Reversibility**  Reversibility means that the positional encoding is an injective map – that is, every coordinate is mapped to a unique rotation, thus position can be recovered. This property is important in Liu and Zhou [44] and Su [68] to derive Axial RoPE. While it prevents Eq. 11, it is necessary only for the $D = 1$ case. More generally, Mixed RoPE can learn an injective map for large $D$. Moreover, while having a "lossless" positional encoding is nice mathematically, its practical utility has yet to be soundly justified, especially if the positional encoding is learnable.

## F Fast Implementation

We follow a vectorized implementation for Spherical RoPE similar to the "fast implementation" proposed in Su et al. [69].

First, apply the rotation directly on after the other:

$$z_d[1] = \cos(\omega_y p_y)\, z_d[1] - \sin(\omega_y p_y)\, z_d[3] \tag{51}$$

$$z_d[3] = \sin(\omega_y y)\, z_d[1] + \cos(\omega_y)\, z_d[3], \tag{52}$$

then

$$z_d[2] = \cos(\omega_y p_x)\, z_d[2] - \sin(\omega_x p_x)\, z_i[3] \tag{53}$$

$$z_d[3] = \sin(\omega_x p_x)\, z_d[2] + \cos(\omega_x p_x)\, z_d[3], \tag{54}$$

where steps 51 and 52 happen simultaneously, and steps 53 and 54 occur at the same time.

## G Experimental Setup

**Models**   We use the ViT-S backbone from the timm library [79]. The network always has a depth of 12. We keep $N$ as close to constant across models as we can. For CIFAR100, the embedding dimensions are changed from $64 \times N_{\text{heads}}$ to $60 \times N_{\text{heads}}$ to be compatible with pairs, triplets and quadruples. For ImageNet, we make the embedding dimension $63 \times N_{\text{heads}}$ for Spherical RoPE and $64 \times N_{\text{heads}}$ for other methods. For classification, we use a class token to pool the tokens and predict. Unlike the patch tokens, the class token is not affected by any positional encoding.

**CIFAR100**   All experiments on CIFAR100 were performed on one A100 GPUs with a batch size 256. We use a patch size of $4 \times 4$ on the original image size $32 \times 32$. The training uses heavy regularization and augmentations including dropout, MixUp [87] and CutMix [86]. The models are trained for 400 epochs, taking $\sim 40$ seconds per training loop.

**ImageNet**   All experiments on ImageNet-1k were performed on four A100 GPUs with a batch size 256. We used cosine learning rate with a learning rate of $3e-3$ for 200 epochs with 5 epochs of linear warm-up. We used a patch size of $16 \times 16$ on the cropped and resized $224 \times 224$ image after applying 3-Augment [72]. We use the LAMB [84] optimizer. All experiments took $\sim$20 hrs with $\sim 5$ to $8$ minutes to complete a training loop depending on method.

**Positional Encodings**   For testing with different resolutions, the images from ImageNet's validation set were normalized, resized and cropped. On training, the patches were assigned position $[-\pi, \pi]$ and for evaluation, the patch positions were extrapolated to the range $[-\frac{P}{P_0}\pi, \frac{P}{P_0}\pi]$. For Learned APE, the positional embeddings are instead interpolated. The fixed frequencies were given by $\omega_d = 1/100^{2d/D}$, where $d$ is the index of the pair/tuple/quadruple. One frequency is shared between both $x$ and $y$ in our implementation of Axial RoPE .

# H  Hyperparameters

Table 5: Hyperparameters for ImageNet-1K Training

| Category | Setting |
|---|---|
| **Model Architecture** | |
| Patch Size | 16x16 |
| Heads | 6 |
| Latent Dimension | 64 (63 for Spherical) $\times$ Heads |
| Depth | 12 |
| Pooling | [CLS] |
| Stochastic Depth | No |
| Dropout | No |
| LayerScale | 1 |
| **Optimization** | |
| Optimizer | LAMB [84] |
| Base Learning Rate | 4e-3 |
| Weight Decay | 0.05 |
| Learning Rate Schedule | Cosine Decay |
| Warmup Schedule | Linear |
| Warmup Epochs | 5 |
| Epochs | 200 |
| Batch Size | 512 |
| Gradient Clipping | ✓ |
| **Precision and Backend** | |
| Precision | Mixed (bfloat16) |
| Backend | torch.autocast |
| **Data Augmentation - Train** | |
| Crop | RandomResizedCrop (192→224) |
| Flip | ✓ |
| 3-Augment | ✓ |
| Color Jitter | (0.3, 0.3, 0.3, 0.0) |
| Mixup [87] | ✗ |
| Cutmix [86] | ✗ |
| Normalization | ImageNet-1K Statistics |
| **Data Augmentation - Test** | |
| Resize | Resize → Resolution |
| Crop | CenterCrop |
| Normalize | ImageNet-1K Statistics |

Table 6: Hyperparameters for CIFAR100 Training

| Category | Setting |
|---|---|
| **Model Architecture** | |
| Patch Size | 16x16 |
| Heads | 12 |
| Latent Dimension | $60 \times$ Heads |
| Depth | 12 |
| Pooling | [CLS] |
| Stochastic Depth | 0.1 |
| Dropout | 0.1 |
| LayerScale | ✓ |
| **Optimization** | |
| Optimizer | LAMB [84] |
| Base Learning Rate | 4e-3 |
| Weight Decay | 0.05 |
| Learning Rate Schedule | Cosine Decay |
| Warmup Schedule | Linear |
| Warmup Epochs | 5 |
| Epochs | 400 |
| Batch Size | 1024 |
| Gradient Clipping | ✓ |
| **Precision and Backend** | |
| Precision | Mixed (bfloat16) |
| Backend | torch.autocast |
| **Data Augmentation - Train** | |
| Crop | RandomResizedCrop (32) |
| Flip | ✓ |
| 3-Augment | ✓ |
| Color Jitter | (0.3, 0.3, 0.3, 0.0) |
| Mixup [87] | 0.8 |
| Cutmix [86] | 1.0 |
| Normalization | CIFAR Statistics |
| **Data Augmentation - Test** | |
| Normalize | CIFAR Statistics |

# I Additional Evaluations

In this section, we include extra evaluations including, basic data scaling, segmentation and speed. We also include additional experiments on the effect of rotation frequencies on Uniform RoPE.

## I.1 Data Scaling

Below we evaluate the data scaling of each method. We partition the CIFAR100 dataset into smaller subsets. The number of epochs is scaled, so that the number of training steps is matched on the data subsets. Each model is trained only once on each data split. This experiment tests whether a commutative constraint is beneficial in smaller data regimes as an inductive bias.

Table 7: Performance on different portions of CIFAR100.

| Dataset Size | Spherical (Learned) | Axial (Learned) | Mixed | Uniform | APE |
|---|---|---|---|---|---|
| 0.2 | 56.04 (**57.2**) | 55.3 (56.6) | 56.9 | 52.82 | 45.9 |
| 0.4 | 63.6 (**65.34**) | 63.3 (62.5) | 64.4 | 59.7 | 53.4 |
| 0.6 | 67.6 (69.8) | 66.0 (66.78) | **70.0** | 64.1 | 57.7 |
| 0.8 | 69.8 (**72.6**) | 69.9 (69.1) | 71.6 | 65.8 | 59.0 |

Equivariance, in theory, should provide better performance at small scales due to its inductive bias. However, we observe that learned Spherical RoPE performs on-par or better than Mixed RoPE with less parameters. The small gap

## I.2 Segmentation

Below we include rudimentary experiments on segmentation to show that the equivalent performance of Spherical RoPE is not caused by the simplicity of classification as a task. For these experiments, we use the models trained on ImageNet-1k as pretrained backbones and fine-tune for Pascal VOC Segmentation [19]. The heads of the models are replaced with a single MLP which is used to get patch logits for each of . Bilinear interpolation is used to create individual pixel logits.

Table 8: Segmentation results (IoU) on VOC with and without augmentation.

| | Spherical | Axial (Learned) | Mixed | Uniform |
|---|---|---|---|---|
| VOC (No Aug.) | **0.45**(0.46) | 0.42 (0.43) | 0.44 | 0.41 |
| VOC (Simple Aug.) | $0.498_{\pm.007}$ (**$0.502_{\pm.012}$**) | $0.474_{\pm.011}$ ($0.468_{\pm.010}$) | **$0.502_{\pm.008}$** | $0.461_{\pm.012}$ |

## I.3 Wall Clock Time

Below we include the wall clock time for each method. Beyond vectorization as described in Appendix F, no optimizations were made for speed. LieRE was implemented following the pseudo-code in Ostmeier et al. [52].

Table 9: Time comparison across different positional encodings

| Time comparison | Spherical (Learned) | Axial (Learned) | Mixed | LieRE | APE | Uniform |
|---|---|---|---|---|---|---|
| Without `torch.autocast` | 16.6s (16.6s) | 16.5s (16.7s) | 15.7s | 27.4s | 13.1s | 16.5s |
| With `torch.autocast` | 6.7s (5.8s) | 6.5s (5.7s) | 5.2s | 13.6s | 3.9s | 6.6s |

The experiment was performed by running a dummy input of dimension (B=256, C=3, H=224, W=224) 100 times with a ViT backbone on one A100 gpu. This is simulated training time, so the rotation matrices were recalculated with every pass for learnable methods.

Note, Mixed RoPE is faster due to naive the use of naive vector partitioning operations and broadcasting. The main conclusion is that learning parameters and Spherical RoPE cause negligible computational overhead.

## I.4 Learned Frequencies

When the frequencies of Spherical RoPE are learned, it is possible for the model to learn equivariance in a particular layer. Like Mixed RoPE, if the rotation frequencies in a layer is set to zero, then the

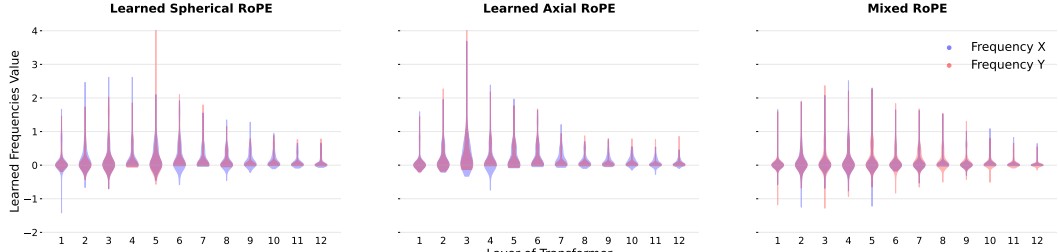

Figure 4: The distribution of learned frequencies in each layer of the ViT. Every method tends to learn low frequency positional encodings in the later layers of the network, meaning representations in the later layers are more invariant to position.

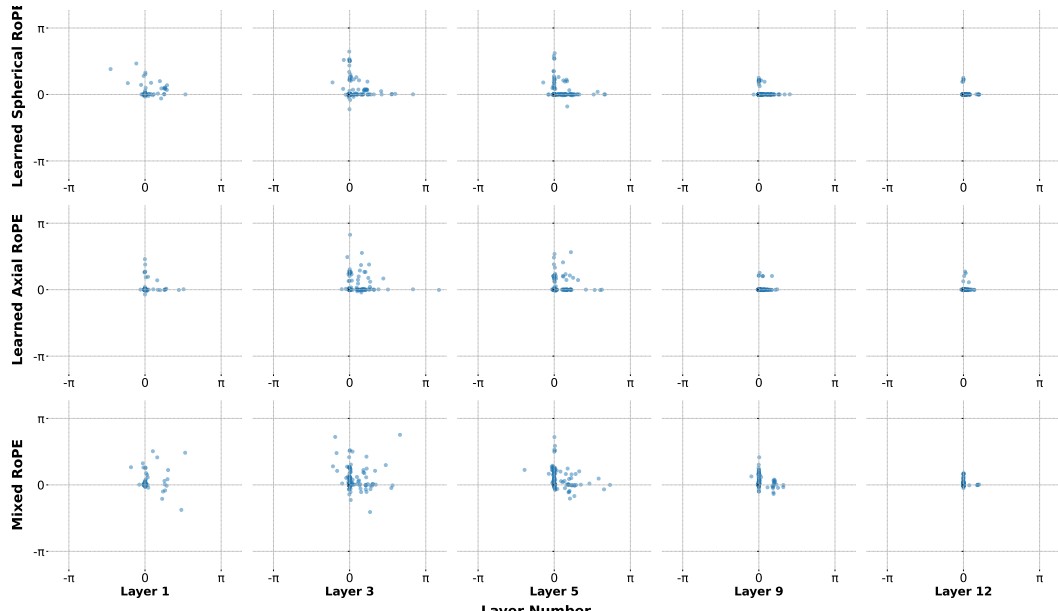

Figure 5: The scatterplot of learned $\omega_x$ and $\omega_y$. Note, though Axial RoPE is plotting $\omega_x$ and $\omega_y$ together, the rotations will always be axial, so there is no importance to the pairing.

attention score is position invariant. If one of the rotation frequencies is set to zero, then Spherical RoPE will become trivially equivariant in the remaining direction. This makes it interesting to observe what weights the model learns. Below we show the learned frequencies in each layer of the network after being trained on ImageNet-1k.

Because frequencies progressively trend toward the axis in deeper layers of the network which makes the positional encodings equivariant in that direction, one could argue that Spherical RoPE learns an equivariant representation in its later layers. However, this same trend can be seen in Mixed RoPE and more notably in Axial RoPE. Because Axial RoPE assumes mutual exclusivity, the frequency pairing is arbitrary. Since we still see the trend toward the axes, the observation that later layers use lower frequencies could be an artifact of backpropagation rather than a necessity for the model to learn an equivariant representation.

Interestingly, every method has notable clusters at zero frequencies. This suggests that much of the information in the images may be position agnostic. This further explains why setting low frequencies to zero in traditional RoPE improves performance as observed in Barbero et al. [5]. An additional cluster can be observed most notably in the later layers of Spherical and Axial RoPE. We hypothesize this frequency corresponds to some information about the resolution of the image, i.e. the spacing of the grid. Some insight on how to generalize to higher resolutions may come from how this frequency corresponds to training data.

# J Proofs and Lemmas

**Axial RoPE Separability**

> **Proposition 3.** *Axial RoPE is separable in $x$ and $y$, that is, the attention score can be decomposed into,*
> $$\alpha(\mathbf{x}_i, \mathbf{x}_j, \mathbf{p}_i, \mathbf{p}_j) = \alpha_{ij}^{(x)} + \alpha_{ij}^{(y)}$$

**Proof.** Suppose we define the dot-product attention score as
$$\alpha(\mathbf{q}, \mathbf{k}) = \mathbf{q}^\top \mathbf{k}.$$

We incorporate *Axial Rotary Positional Embeddings* by rotating each 2-dimensional subvector of the query (and likewise the key). Concretely, if the hidden dimension is $2n$, we partition
$$\mathbf{q} = \begin{bmatrix} \mathbf{q}_{x,1}, \mathbf{q}_{y,1}, \ldots, \mathbf{q}_{x,n}, \mathbf{q}_{y,n} \end{bmatrix}^\top, \quad \mathbf{k} = \begin{bmatrix} \mathbf{k}_{x,1}, \mathbf{k}_{y,1}, \ldots, \mathbf{k}_{x,n}, \mathbf{k}_{y,n} \end{bmatrix}^\top, \tag{55}$$

where each $\mathbf{q}_{x,d}, \mathbf{q}_{y,d}, \mathbf{k}_{x,d}, \mathbf{k}_{y,d} \in \mathbb{R}^2$. At spatial location $\mathbf{p} = (p_x, p_y)$, we apply rotations
$$\mathbf{q}'_{x,d} = \mathbf{R}(\omega_d\, p_x)\, \mathbf{q}_{x,d}, \quad \mathbf{q}'_{y,d} = \mathbf{R}(\omega_d\, p_y)\, \mathbf{q}_{y,d},$$

and similarly for $\mathbf{k}$. Here $\mathbf{R}(\theta) \in \mathbb{R}^{2 \times 2}$ is the planar rotation by angle $\theta$.
For tokens at positions $\mathbf{p}_i = (p_{i,x}, p_{i,y})$ and $\mathbf{p}_j = (p_{j,x}, p_{j,y})$, their rotated queries and keys yield
$$\alpha_{ij} = \sum_{d=1}^{n} \Big[ (\mathbf{q}_{x,d})^\top \mathbf{R}(\omega_d\,(p_{j,x} - p_{i,x}))\, \mathbf{k}_{x,d} + (\mathbf{q}_{y,d})^\top \mathbf{R}(\omega_d\,(p_{j,y} - p_{i,y}))\, \mathbf{k}_{y,d} \Big].$$

Define the horizontal and vertical components by
$$\alpha_{ij}^{(x)} := \sum_{d=1}^{n} (\mathbf{q}_{x,d})^\top \mathbf{R}(\omega_d\,(p_{j,x} - p_{i,x}))\, \mathbf{k}_{x,d}, \quad \alpha_{ij}^{(y)} := \sum_{d=1}^{n} (\mathbf{q}_{y,d})^\top \mathbf{R}(\omega_d\,(p_{j,y} - p_{i,y}))\, \mathbf{k}_{y,d}.$$

Hence the total attention decomposes additively:
$$\alpha_{ij} = \alpha_{ij}^{(x)} + \alpha_{ij}^{(y)},$$

demonstrating that *axial* rotary embeddings factorize the positional dependence along each axis. $\quad \square$

**Matrix Exponentiation**  Computing the matrix exponential by exponentiating the eigenvalues is a common result in linear algebra and numerics, however we provide it here for those unfamiliar.

> **Lemma 1.** *Let $\mathbf{A}$ be a diagonalizable matrix $\mathbf{A} = \mathbf{U}\mathbf{\Lambda}\mathbf{U}^{-1}$, then the matrix exponential of $\mathbf{A}$ is given by*
> $$\exp(\mathbf{A}) = \mathbf{U} \exp(\mathbf{\Lambda})\, \mathbf{U}^{-1}$$

**Proof.**
Recall the power-series definition of the matrix exponential:
$$\exp(\mathbf{A}) = \sum_{k=0}^{\infty} \frac{1}{k!} \mathbf{A}^k. \tag{56}$$

Since $\mathbf{A}$ is diagonalizable,
$$\mathbf{A}^k = \left( \mathbf{U}\, \mathbf{\Lambda}\, \mathbf{U}^{-1} \right)^k = \mathbf{U}\, \mathbf{\Lambda}^k\, \mathbf{U}^{-1}. \tag{57}$$

Substituting into the series gives
$$\exp(\mathbf{A}) = \sum_{k=0}^{\infty} \frac{1}{k!} \left( \mathbf{U}\, \mathbf{\Lambda}^k\, \mathbf{U}^{-1} \right) = \mathbf{U} \Big( \sum_{k=0}^{\infty} \frac{1}{k!} \mathbf{\Lambda}^k \Big) \mathbf{U}^{-1}. \tag{58}$$

Because $\mathbf{\Lambda}$ is diagonal, the series $\sum_{k=0}^{\infty} \frac{1}{k!} \mathbf{\Lambda}^k$ is itself the diagonal matrix of scalar exponentials,

$$\exp(\mathbf{\Lambda}) = \mathrm{diag}(e^{\lambda_1}, \ldots, e^{\lambda_n}). \tag{59}$$

Hence is well defined, and

$$\exp(\mathbf{A}) = \mathbf{U} \exp(\mathbf{\Lambda}) \, \mathbf{U}^{-1}. \tag{60}$$

$\square$

**Simultaneous-Diagonalizability**  The proof that two (diagonalizable) matrixes are simultaneous-diagonalizability if and only if they are commutative is also a standard result. However, we once again provide it here:

> **Lemma 2.** *Let $\mathcal{A}_x$ and $\mathcal{A}_y$ be skew-symmetric. Then $\mathcal{A}_x$ and $\mathcal{A}_y$ are simultaneously diagonalizable if and only if $\mathcal{A}_x \mathcal{A}_y = \mathcal{A}_y \mathcal{A}_x$.*

**Proof.**
Suppose $\mathcal{A}_x$ and $\mathcal{A}_y$ are simultaneously diagonalizable. Then, because they are skew-symmetric, there exists a unitary matrix $\mathbf{U}$ such that

$$\mathbf{U}\Lambda_x\mathbf{U}^\top = \mathcal{A}_x \quad \text{and} \quad \mathbf{U}\Lambda_y\mathbf{U}^\top = \mathcal{A}_y, \tag{61}$$

where $\Lambda_x$ and $\Lambda_y$ are diagonal matrices.
Then,

$$\mathcal{A}_x\mathcal{A}_y = \mathbf{U}\Lambda_x\mathbf{U}^\top\mathbf{U}\Lambda_y\mathbf{U}^\top = \mathbf{U}\Lambda_x\Lambda_y\mathbf{U}^\top = \mathbf{U}\Lambda_y\Lambda_x\mathbf{U}^\top = \mathcal{A}_x\mathcal{A}_y \tag{62}$$

Hence, $\mathcal{A}_x$ and $\mathcal{A}_y$ commute.
Now suppose $\mathcal{A}_x$ and $\mathcal{A}_y$ commute, $\mathcal{A}_x\mathcal{A}_y = \mathcal{A}_y\mathcal{A}_x$. Since $\mathcal{A}_x$ and $\mathcal{A}_y$ are skew-symmetric, they are diagonalizable in $\mathbb{C}^{D \times D}$, thus there exists a basis of eigenvectors of $\mathcal{A}_x$. Because $\mathcal{A}_y$ commutes with $\mathcal{A}_x$, the eigenspaces of $\mathcal{A}_x$ are invariant under $\mathcal{A}_y$. That is, for any eigenvalue $\lambda$ of $\mathcal{A}_x$, the corresponding eigenspace

$$E_\lambda = \{v \in \mathbb{C}^D : \mathcal{A}_x v = \lambda v\} \tag{63}$$

is $\mathcal{A}_y$-invariant: if $v \in E_\lambda$, then

$$\mathcal{A}_x(\mathcal{A}_y v) = \mathcal{A}_y(\mathcal{A}_x v) = \mathcal{A}_y(\lambda v) = \lambda \mathcal{A}_y v \Rightarrow \mathcal{A}_y v \in E_\lambda. \tag{64}$$

Now, restrict $\mathcal{A}_x$ to each eigenspace $E_\lambda$. Since $\mathbb{C}$ is algebraically closed and $\mathcal{A}_y|_{E_\lambda}$ is a linear operator on a finite-dimensional space, $\mathcal{A}_y$ is diagonalizable on $E_\lambda$. Thus, we can choose a basis of eigenvectors for $\mathcal{A}_y$ in each $E_\lambda$.
Putting these together, we get a basis for $\mathbb{C}^N$ consisting of vectors that are eigenvectors for both $\mathcal{A}_x$ and $\mathcal{A}_y$. Therefore, $\mathcal{A}_x$ and $\mathcal{A}_y$ are simultaneously diagonalizable.

$\square$

**1-D LieRE is equivalent to RoPE**  In this section, we will more formally prove that the traditional RoPE with learned rotation frequencies is equivalent to 1-D RoPE as proposed in Section 3.

> **Proposition 1.** *Any $D$-dimensional rotation can be parameterized by RoPE with learned frequencies.*

**Proof.**
We define a rotation to be an orthogonal matrix with positive determinant; that is, it is an element of $\mathbf{R} \in \mathrm{SO}(N)$. We can write any element of $\mathrm{SO}(N)$ via the exponential map $\mathbf{R} = e^{\mathcal{A}}$ where $\mathcal{A} \in \mathfrak{so}(N)$, i.e. $\mathcal{A}$ is a skew-symmetric matrix. It is well-known that the eigenvalues of a real, skew-symmetric matrix are purely imaginary (or zero), and such a matrix is unitarily (i.e. orthogonally) diagonalizable over $\mathbb{C}$, resulting in a spectral decomposition with a purely imaginary eigenvalue matrix. Thus,

$$\mathcal{A} = \mathbf{U}\mathbf{\Lambda}i\mathbf{U}^\dagger \tag{65}$$

and, by Lemma 1,

$$\exp(\mathcal{A}) = \mathbf{U} \exp(\mathbf{\Lambda}i) \, \mathbf{U}^\dagger. \tag{66}$$

where, because $\mathbf{\Lambda}$ is diagonal, $\exp(\mathbf{\Lambda})$ is simply the scalar-exponential of each element. The positional encoding of a token to a query can be written as,

$$\varphi(\mathbf{x}, p) = \exp(\mathcal{A}p)\mathbf{W}_q\mathbf{x} = \mathbf{U}\exp(\mathbf{\Lambda}i\,p)\mathbf{W}_\mathbf{q}'\mathbf{x} \tag{67}$$

where $\mathbf{W}'_q = \mathbf{W}_q \mathbf{U}$. We assume the same encoding for the key with a different matrix, $\mathbf{W}'_k$ and the same generator, $\mathcal{A}$. This equation can be rewritten as $\varphi(\mathbf{x}, p) = \mathbf{U}RoPE(\mathbf{x}, p)$ by Eq.10. If the attention score is given by $\alpha(\mathbf{q}, \mathbf{k}) = \mathbf{q}^\dagger \mathbf{k}$, where $\dagger$ denotes the Hermitian transpose, then the attention score can be expanded into,

$$\alpha(\mathbf{x}_i, \mathbf{x}_j, p_i, p_j) = RoPE(\mathbf{x}_i, p_i)^\dagger \mathbf{U}^\dagger \mathbf{U} RoPE(\mathbf{x}_j, p_j) \tag{68}$$

$$= RoPE(\mathbf{x}_i, p_i)^\dagger RoPE(\mathbf{x}_j, p_j). \tag{69}$$

Hence, *any LieRE of one generator can be expressed as RoPE with learned rotation frequencies.* $\square$

**Any commutative LieRE is equivalent to Mixed RoPE**   We now prove that multi-dimensional LieRE with commutative generators generalizes directly to Mixed RoPE.

---

**Proposition 2.** *Any $M$-dimensional LieRE with commutative generators can be parameterized by Mixed RoPE.*

---

**Proof.**
Let $\mathcal{A}_1, \ldots, \mathcal{A}_M \subset \mathfrak{so}(N)$ be skew-symmetric generators such that $[\mathcal{A}_m, \mathcal{A}_n] = \mathbf{0}$ for all $m, n$. By Lemma 2, commuting normal matrices are simultaneously unitarily diagonalizable. Thus, there exists a unitary $\mathbf{U}$ and diagonal matrices $\mathbf{\Lambda}_1, \ldots, \mathbf{\Lambda}_M$ such that

$$\mathcal{A}_m = \mathbf{U}\mathbf{\Lambda}_m i \mathbf{U}^\dagger \quad \text{for all } m = 1, \ldots, M. \tag{70}$$

For a position vector $\mathbf{p} = (p_1, \ldots, p_M) \in \mathbb{R}^M$, the LieRE positional encoding is

$$\text{LieRE}(\mathbf{x}, \mathbf{p}) = \exp\left(\sum_{m=1}^{M} \mathcal{A}_m p_m\right) \mathbf{W} q \mathbf{x}, \tag{71}$$

which, using Lemmas 1 and 2, can be written as

$$\text{LieRE}(\mathbf{x}, \mathbf{p}) = \mathbf{U} \exp\left(\sum_{m=1}^{M} \mathbf{\Lambda}_m i, p_m\right) \mathbf{U}^\dagger \mathbf{W}_q \mathbf{x}. \tag{72}$$

Let $\mathbf{W}'_q = \mathbf{U}^\dagger \mathbf{W}_q$. Then

$$\text{LieRE}(\mathbf{x}, \mathbf{p}) = \mathbf{U}\text{MixedRoPE}(\mathbf{x}, \mathbf{p}), \tag{73}$$

where MixedRoPE applies elementwise complex rotations

$$e^{i(\lambda_1^{(k)} p_1 + \cdots + \lambda_M^{(k)} p_M)} \tag{74}$$

to each channel $k$, with frequencies $\lambda_m^{(k)}$ learned from $\mathbf{\Lambda}_m$.
If the attention score is given by $\alpha(\mathbf{q}, \mathbf{k}) = \mathbf{q}^\dagger \mathbf{k}$, then

$$\alpha(\mathbf{x}_i, \mathbf{x}_j, \mathbf{p}_i, \mathbf{p}_j) = \text{MixedRoPE}(\mathbf{x}_i, \mathbf{p}_i)^\dagger \mathbf{U}^\dagger \mathbf{U}\text{MixedRoPE}(\mathbf{x}_j, \mathbf{p}_j) \tag{75}$$

$$= \text{MixedRoPE}(\mathbf{x}_i, \mathbf{p}_i)^\dagger \text{MixedRoPE}(\mathbf{x}_j \mathbf{p}_j). \tag{76}$$

Hence, *any $M$-dimensional LieRE with commutative generators is equivalent to a Mixed RoPE parameterization with learned rotation frequencies.* $\square$

