# OpenReview forum: "A Circular Argument: Does RoPE need to be Equivariant for Vision?"
_NeurIPS.cc/2025/Conference — NeurIPS 2025 poster_

### Official Review · Reviewer_5p81 · 2025-06-10

**Clarity:** 4
**Significance:** 2
**Originality:** 3
**Rating:** 4
**Confidence:** 4

**Summary:**

This paper investigates the role of equivariance in Rotary Positional Encodings (RoPE) for vision transformers. The authors formulate a unifying theoretical framework for various multi-dimensional RoPE extensions (such as Axial RoPE, Mixed RoPE, and LieRE) and formally show that Mixed RoPE is the most general form of equivariant rotary encoding under the assumption of commuting generators (a necessary condition for equivariance). The paper questions the belief that this equivariance (i.e., being a relative positional encoding) is the key to RoPE's success. To test this hypothesis, the authors propose "Spherical RoPE", a non-equivariant RoPE variant based on anti-commuting generators (SO(3) rotations), designed as a minimal modification to equivariant counterparts. Through experiments on CIFAR-100 and ImageNet-1K, they demonstrate that Spherical RoPE performs on par with equivariant RoPE variants. The authors conclude that properties other than equivariance, such as the diversity of rotation frequencies, might be more critical for the performance of rotary embeddings in vision tasks.

**Questions:**

1. Have the authors tried to interpret what kind of positional relations architectures with "unrestricted" RoPE variants (i.e. not enforcing translational equivariance) learn? For example, this could be analysed by looking at the learned attention maps for different (shifted) versions of the same image: If the attention maps transform (shift) in a similar manner, it would indicate that models have learned approximate translational equivariance. If not, it would reveal that such models can learn attention patterns that are (by construction) inaccessible to architectures that use translationally equivariant RoPE variants. Both outcomes would be interesting.
2. The authors should report error bars on the results, so that their statistical significance can be assessed. Ideally, they should extend the experimental evaluation to include other vision tasks (e.g. segmentation) and architectures to see whether the findings hold in general.
3. The authors should re-run experiments for LieRE using their training framework, so that the different results in Table 2 can be compared more directly.
4. In Figure 3, it seems that the highest ImageNet top1-accuracy is obtained for image resolutions *above* the training resolution for all tested RoPE variants. This feels counterintuitive to me (I would have expected that performance peaks for the training resolution), can the authors provide any insights on potential explanations for this trend?
5. The paper uses the term "equivariant" synonymous with "translation equivariant". However, 2D positional encodings can also be *rotation* equivariant. This means that for a relative displacement $p_{ij}$, the attention pattern is the same no matter in which direction $p_{ij}$ points, only the length of $p_{ij}$ would be important (In Figure 1A, this would correspond to a circular attention pattern). I believe neither of the discussed RoPE variants has this property, but it is conceivable that this could be advantageous for some vision tasks (e.g. classifying objects on a table from a bird's eye perspective, see https://arxiv.org/abs/1612.08498 and similar works for context). I believe it would be valuable to discuss the distinction between translation and rotation equivariance in the manuscript.



Typos and other minor formatting issues (these do not influence the rating):

- "effects" on line 87 should probably be "affects" (see also the caption of Figure 2, where there are similar mixups)
- inline equation on line 91 uses $p$, but the text on line 92 refers to $p_t$
- $_1^{(y)}$ on line 140 should probably be $z_1^{(y)}$
- "guage" on line 253 should be "gauge"
- "that" on line 286 should be "than"
- "method" on line 302 should be "methods"

**Ethical Concerns:**

["NO or VERY MINOR ethics concerns only"]

**Final Justification:**

The authors ran some additional experiments, answered my questions, and improved the discussion/interpretation of the results.

**Limitations:**

yes

**Paper Formatting Concerns:**

Point 8 (about compute resources) in the paper checklist is still marked as TODO. However, I think the paper already provides sufficient information in this regard (experiments seem to have been run on A100s, I don't think more information is required).

**Quality:**

2

**Strengths And Weaknesses:**

**Strengths**
1. The paper challenges the widely-held assumption that the success of RoPE stems from being relative (translational equivariance) and provides empirical evidence that built-in equivariance may not be a requirement for successful positional encodings in vision applications.
2. The paper offers a valuable theoretical clarification by unifying various RoPE extensions within a common framework.
3. The proposed "Spherical RoPE" is an elegant way to test the central hypothesis of the paper.
4. The paper is well-written and the presented arguments are easy to follow.

**Weaknesses**
1. The authors find that built-in ("hardcoded") translational equivariance may not be necessary for positional encodings, but they do not analyse what the models with "unrestricted" RoPE variants actually learn. It is conceivable that those models still learn approximate ("good enough") translational equivariance.
2. The presented results seem to be from a single run and do not report any error bars. For the main performance comparison in Table 2, an indication of statistical significance seems necessary to arrive at the conclusion that performance truly is "equivalent". Similarly, for the extrapolation experiment in Figure 3, the statistical significance of the results is unclear and care should be taken when drawing conclusions such as "Spherical RoPE generalizes better to larger image sizes compared to other RoPE variants".
3. The experiments are limited in scope; they were performed only for a single architecture (ViT-S), a single task (classification) and on two datasets (CIFAR100 and ImageNet-1K). It is conceivable to me that the benefits of translational equivariance only become apparent in different (perhaps larger?) architectures, or on different tasks like object detection or segmentation, which might rely more heavily on spatial relationships. The strong conclusion that equivariance is not needed for vision (in general) is not fully supported by the limited experimental evidence.
4. The authors compare their methods to LieRE using results from the original LieRE paper, acknowledging that the training protocol is different. This makes a direct comparison of performance numbers difficult. While the authors' main conclusions do not hinge on these results, a re-implementation of LieRE within their own experimental framework would allow a more direct comparison and seems to be straightforward.
5. While the authors' findings are certainly interesting, their impact on the field seems limited to me. While relative positional embeddings may not be required and similar performance can be reached without translational equivariance, as long as there is no advantage to be gained from dropping this constraint (either increased performance or efficiency), it is not clear why this insight is useful.

---

> ### Author Rebuttal · Authors · 2025-07-31
>
> # Organization
>
> Because there seemed to be consistent confusion about the goal and significance about our work we have included a blanket statement at the end, after the individual responses. We have also included the additional results in that section as it was a consistent concern across reviews. We will provide the remaining numbers as soon as possible (expect Saturday for ImageNet).
>
> ---
>
>
> **Weaknesses**
>
> >The authors find that built-in ...
>
> It is a good question, though to be pedantic, Spherical RoPE is not unconstrained; it is contrained to *not* be commutative. It is true that the models learn to be almost completely translation equivariant (see the Questions section for elaboration).
>
> While it is an interesting question, we don't think it is relevant enough to cram into the main body of this paper. The point still stands that the hard constraint does not appear to be why RoPE performs well.
>
> >The presented results seem to be from ...
>
> We have included additional experiments below.
>
> >Similarly, for the extrapolation experiment in Figure 3, the statistical significance of the results is unclear and care should be taken when drawing conclusions such as "Spherical RoPE generalizes better to larger image sizes compared to other RoPE variants".
>
> We will include results from three runs when those results are available (expect Saturday/Sunday).
>
> >The experiments are limited in scope; ...
>
> Our claim is that ***RoPE*** does not need to be strictly equivariant, that is the benefits from RoPE are not caused by equivariance. We would like to make it clear that we are not claiming that "equivariance is not needed for vision (in general)."
>
> We have included segmentation experiments on VOC Segmentation, however this tends to require a pretrained backbone for transformer based architectures to have good results. Even in this very low data regime, we see that Spherical RoPE performs on par which is contrary to the main motivation of shift invariance as an inductive bias.
>
> > The authors compare their methods ...
>
> This has now been included.
>
>
> >While the authors' findings ...
>
> RoPE is currently the most popular way of doing positional encodings and the question of how to extend it to N-D is a very relevant and active topic right now. It is very relevant for the field to understand why RoPE works to move forward. It is currently assumed that it is because it is a relative positional encoding (see blanket statement).
>
>
> Questions:
>
> >Have the authors tried to interpret ...
>
> Yes we have. The Learned Spherical RoPE learns to push all the $y$ frequencies to zero in the last layers which collapes the rotation to the identity. This is a special case where Spherical RoPE becomes strictly equivariant to $x$. We will try to provide a small section in the appendix because it *is* interesting.
>
> However, this requires more testing such as whether this is determined by the order of the roll/yaw matrix. Moreover, whether it learns to be approximately equivariant or not isn't relevant for this paper's focus question of whether it is *necessary*. Indeed, it almost needs to learn approximate invariance for these datasets since we do random cropping augmentations. Because of the history and perceived importance of equivariance in these datasets, our results are even more evidence that equivariance is not the core reason RoPE works.
>
>
> >The authors should report error bars on the results...
>
> We have provided these results below, though some results are in progress.
>
> >The authors should re-run experiments for LieRE using their training framework, so that the different results in Table 2 can be compared more directly.
>
> CIFAR100 provided. ImageNet in progress.
>
> >In Figure 3, it seems that the highest ImageNet ...
>
> This is a standard result and one of the draws of RoPE-like embeddings as it allows for training on lower resolution images which can be done much faster [1-5]. We are unfamiliar if there is any proper explanation of this phenomena, but the layman intuition is that the larger resolution provides more information and more patches as context for the classification. The change in zoom in the content of the patches is not an issue because of random cropping while training.
>
> >The paper uses the term "equivariant" ...
>
> Thanks for pointing this out. We discuss this concept in detail in our discussion of equivariance in the Appendix, but we have now added a sentence stating that we assume equivariance -> shift equivariant explicitly in the main body.
>
> We have thought about shift equivariance. To make a rotationally equivariant RoPE, one simply needs to parameterized position in log-polar coordinates. However, SE(2) is much more complicated. One needs a position and orientation [6] of the patch and the orientation is not a known quantity. More generally, one could construct a bi-invariant, which has been shown to be equivalent to the steerability property in [7].
>
> [6] Kofinas, M., Nagaraja, N., & Gavves, E. (2021). **Roto-translated local coordinate frames for interacting dynamical systems.** Advances in Neural Information Processing Systems, 34, 6417-6429.
>
> [7] Knigge, D., Wessels, D., Valperga, R., Papa, S., Sonke, J. J., Bekkers, E., & Gavves, E. (2024). **Space-time continuous pde forecasting using equivariant neural fields.** Advances in Neural Information Processing Systems, 37, 76553-76577.
>
> >Typos and other minor formatting issues (these do not influence the rating):
> >
>
> Thank you for the diligence!
>
> ---
>
>
> # Blanket Statement
>
> ----
> ## Significance
> As we feel like several reviewers may have misunderstood the goal of our work, we would like to clarify on this point: Our goal is understand why RoPE works, and specifically whether equivariance is the dominant contributer as is commonly believed. Our claim is *not* that one should use Spherical RoPE or Uniform RoPE. These are ablations to move towards understanding what is important for RoPE's success.
>
> Because RoPE has been incorporated into every large model, it is a very relevant question for this year to understand why exactly it works. It is strongly believed that the improvement is because RoPE is a relative positional encoding. We believe we clearly show that this is strictly not true. Whether Liere, Mixed RoPE, or Spherical RoPE perform *slightly* better is not relevant as the understanding is RoPE is better because it is equivariant. As Spherical RoPE (which is not relative) maintains its gap over APE where Uniform and Axial RoPE (which are relative) has a weakened performance, equivariance does not seem *necessary* for the main benefits of RoPE.
>
> We believe that this is significant in context of *very* recent work. Many contemporary works seek to impose constraints, such as commutativity, to "extend" LieRE. We show by Proposition 2 that if we need to create a rotary embedding which is commutative, then whatever we do will theoretically boil down to Mixed RoPE. We also show through experiments that reversibility [3] (Axial RoPE) produces worse performance. To illustrate the impact, we point the reviewers to three recent papers doing exactly this that have been accepted or submitted in the last three months:
>
> [1] Connor Schenck, Isaac Reid, Mithun George Jacob, Alex Bewley, Joshua Ainslie, David Rendleman, Deepali Jain, Mohit Sharma, Avinava Dubey, Ayzaan Wahid, et al.
> **Learning the ropes: Better 2d and 3d position encodings with string.** (ICML July 2025 Spotlight Poster)
>
> [2] Yu, H., Jiang, T., Jia, S., Yan, S., Liu, S., Qian, H., ... & Yuan, C. (2025). **ComRoPE: Scalable and Robust Rotary Position Embedding Parameterized by Trainable Commuting Angle Matrices.** In Proceedings of the Computer Vision and Pattern Recognition Conference (pp. 4508-4517).
>
> [3] Liu, H., & Zhou, H. (2025). **Rethinking RoPE: A Mathematical Blueprint for N-dimensional Positional Encoding.** (Preprint)
>
> ---
> ## Additional Experiments
>
> We will add results when they are available.
>
> We improved the frequency initialization for learnable RoPEs and have run these additional evaluations:
>
> |  Accuracy with STDev | Spherical (*Learned*)   | Axial  (*Learned*)   | Mixed | Uniform | APE| LieRE|
> | --------           | --------     | -------- | -------- |-------- | -------- | -------- |
> | CIFAR100 (5 Runs)  | 73.2 +- .4 (*74.1 +- .4*)| 72.1+-0.6 (*72.9 +- .6*)|**74.7+-.3**| 70.5+-.2 |64.2 +- 0.9|74.2
> |ImageNet (In Progress) |||
>
> | Segmentation (IoU) | Spherical | Axial (Learned)    | Mixed | Uniform | APE |
> | --------           | --------     | -------- | -------- |-------- |-------- |
> | VOC  (No Augmentation)              |     .45     | .42     | .41|
> |VOC (Simple Augmentation)|.50|.46 (.47)|.50|.45||
>
> | Portion of CIFAR100 | Spherical (Learned) | Axial (Learned) | Mixed | Uniform | APE |
> | -------- | -------- | --------   |------ | ------ |------ |
> | .2     | 56.04 (**57.2**)    | 55.3 (56.6)       | 56.9 | 52.82 |45.9
> | .4     | 63.6  **(65.34)**   |   63.3 (62.5) | 64.4 | 59.7 | 53.4
> | .6     | 67.6  (69.8)   |   66.0 (66.78)    | **70.0** | 64.1| 57.7
> | .8     | 69.8     | 69.9     |**71.6**| |59.0|
>
> ----
>
> Below we show the time it took for each method to run 100 forward passes to compare relative speed. The experiment was performed by running a dummy input of dimension (B=256, C=3, H=224, W=224) 100 times with a ViT backbone on one A100 gpu. This is simulated training time, so the rotation matrices were recalculated with every pass.
>
> | Time comparison | Spherical (Learned) | Axial (Learned)| Mixed | LieRE| APE | Uniform |
> | -------- | -------- | -------- | -------- |-------- |-------- |-------- |
> | Without torch.autocast     | 16.6s  (16.6s)| 16.5s (16.7s)   | 15.7s  | 27.4s| 13.1s | 16.5s
> |With torch.autocast| 6.7s (5.8s)|6.5s (5.7s)| 5.2s | 13.6s| 3.9s |6.6s

---

> > ### Comment · Reviewer_5p81 · 2025-08-01
> >
> > I feel like the blanket statement provided by the authors may primarily address the criticism of other reviewers, the main goal of the paper was quite clear to me (and I believe my criticism and questions all revolve around this main goal).
> >
> > Note that for weakness 3, I understand that you are not claiming that "equivariance is not needed for vision (in general)." However, it is still conceivable that the built-in equivariance of standard RoPE is important for good performance in some other vision task. I merely wanted to point out that the conclusion that equivariance in RoPE is non-essential may not apply to all vision tasks, since you mainly investigated classification tasks. It is good that you now at least also include some experiments on segmentation, which I think makes the conclusion stronger.
> >
> > I think this is good work with interesting results, but I believe the discussion and interpretation of the results should be more nuanced. For example, the authors seem to confirm my suspicion that the alternative RoPE formulations still learn translational equivariance, even though this property is not built-in. My interpretation of this result is that translational equivariance *is* important, it merely means that "hardcoding" equivariance is not strictly necessary for good performance (the model can also learn it by itself). However, this is still an important aspect to consider when thinking about extensions to N-D: While it may not be necessary to hardcode equivariance in higher-dimensional extensions, it *may* be required that translation equivariance is at least *representable* (so that the model is able to learn it). Also, I think it is very natural to ask the question: "Why should we drop exact translation equivariance if the model needs to learn it anyway?" Maybe it makes an extension to N-D easier, but maybe it is also much harder to learn this property in higher dimensions from data alone. Of course, this is all very speculative, which is exactly why I think a nuanced discussion is required and helpful.

---

> ### Author Response · Authors · 2025-08-04
> **Adding Nuance**
>
> Thank you for the clarification.
>
> Yes, you are correct. It is very important for the model to be able to represent equivariance. We have observed in follow up work that making it *impossible* for the model to learn relative position is severely detrimental to the performance. If the query and the key are rotated in opposite directions, then the model can only represent the *mean* position rather than the *relative* position. Our rudimentary experiments observed that this formulation performs as badly as no positional encodings.
>
> As for why we should drop the strict requirements if it needs to learn it anyways: Unlike many N-D systems in science which are perfectly equivariance, vision is not strictly equivariant but only approximately equivariant. This was shown in [1] where softened weight sharing gives better results than strict equivariance. This makes vision an interesting task for assessing positional encodings. Furthermore, we focus on vision because it is the standard benchmark in all related N-D RoPE extensions.
>
> We revised the paper with the below changes to try to weaken the claim and add nuance. I hope this addresses your concerns on the nuance.
>
> ---
>
> 1. Abstract :
> >Line 9-10 : "However, we question the necessity of equivariance." -> "However, we question [whether strict equivariance plays a large role in RoPE's performance]."
>
> 2. Introduction:
> >Line 67-68 : "We conclude that there is evidence that a rotary positional encoding need not be equivariant for vision." -> "We conclude that [our evidence points to the performance of RoPE over traditional embeddings not being explained by equivariance]."
>
> 3. Table 1 :
> >"Equivariant" -> "Strictly Equivariant"
>
> 4. Spherical RoPE:
> > Line 237: In fact, their generators are strictly *non-commutative*, $\mathcal{A}_x\mathcal{A}_y \ne\mathcal{A}_y\mathcal{A}_x$.
> >
> ->
> >In fact, their generators are strictly *non-commutative*, $\mathcal{A}_x\mathcal{A}_y \ne\mathcal{A}_y\mathcal{A}_x$. [While this does not mean Spherical RoPE is incapable of learning or approximating equivariance throughout the network, it is the component of LieRE removed by Mixed RoPE and works which enforce commutativity such as [1] and [2].]
>
> 5. Conclusion:
> >Line 324 : We conclude that Mixed RoPE is a very general solution for M -D data if equivariance is a necessity. However, we see little evidence that relative positional bias is impactful for vision transformers.Thus, we conclude that we do not need equivariance for rotary-based positional encodings to have a performance boost over APE methods."
>
> ->
> >"We conclude that Mixed RoPE is a very general solution for $M$-D data if equivariance is a necessity. However, we see little evidence that strict relative positional bias is impactful for vision transformers. Thus, we conclude that [evidence suggests that RoPE does not *need* stict equivariance constraints to] boost performance over APE methods."
>
> 5. Limitations : Moved to main body.
> >"While our results do not show relative embeddings to be detrimental, we believe them to be evidence that equivariance is not the reason for RoPE's success.
> However, our experiments were performed in Vision where the number of tokens is limited compared to the long context lengths of NLP. Moreover, the datasets are not what many believe to be "at scale". While Spherical RoPE and LieRE would intuitively favored at scale over Axial RoPE, as they have less inductive bias, it is unclear whether inductive bias and equivariance is favored at scale \cite{brehmer2024does}.
> >
> >[It has also been shown that vision is \textit{not} a purely equivariant task and benefits from symmetry breaking [1]. Our results do not show that equivariance is not useful in tasks that are grounded in physics and obey strict symmetries.]"

---

### Official Review · Reviewer_Kc7V · 2025-07-02

**Clarity:** 4
**Significance:** 2
**Originality:** 4
**Rating:** 3
**Confidence:** 3

**Summary:**

This paper investigates whether shift equivariance is really the key factor of success for RoPE in transformers. The authors analyze recent works such as Mixed RoPE, LieRE, Axial RoPE and discuss how commutative generators may not be necessary. They introduce Spherical RoPE (non-commutativity) and Uniform-frequency RoPE to test their hypotheses and empirically show that Spherical RoPE performs similarly to equivariant positional embeddings.

**Questions:**

See weaknesses.

**Ethical Concerns:**

["NO or VERY MINOR ethics concerns only"]

**Limitations:**

See weaknesses.

**Quality:**

3

**Strengths And Weaknesses:**

## Strengths
- This paper tackles the interesting problem of why RoPE is successful and provides theoretical analyses and empirical results to support their claims.
- Propositions 1 and 2 gives good insights into the relationship between RoPE, Mixed RoPE and LieRE. In particular the authors show that Mixed RoPE is the most general equivariant PE.
- The paper is very well written and the presentation is straightforward.

## Weaknesses
- I think the main weakness of this paper is its limited evaluation and experiments. While the theoretical contributions are solid, only the smallest ViT-S model and only two datasets CIFAR100 and ImageNet was used. As the main claim of this paper hinges on the fact that Spherical RoPE performs similarly to Mixed RoPE, the conclusion ("equivariance is not necessary") requires more thorough evaluation at multiple model sizes and with various other datasets.
- A smaller weakness of this paper is that although the authors show that equivariance is not necessary for performance, there is no explanation or hypothesis as to why this may be true. This would greatly strengthen the paper in my opinion.

---

> ### Author Rebuttal · Authors · 2025-07-31
>
> # Organization
>
> Because there seemed to be consistent confusion about the goal and significance about our work we have included a blanket statement at the end, after the individual responses. We have also included the additional results in that section as it was a consistent concern across reviews. We will provide the remaining numbers as soon as possible (expect Saturday for ImageNet).
>
> ---
>
> Reviewer 3 :
> ----
>
> ## Weaknesses
>
> > I think the main weakness of this paper is its limited evaluation and experiments. While the theoretical contributions are solid, only the smallest ViT-S model and only two datasets CIFAR100 and ImageNet was used. As the main claim of this paper hinges on the fact that Spherical RoPE performs similarly to Mixed RoPE, the conclusion ("equivariance is not necessary") requires more thorough evaluation at multiple model sizes and with various other datasets.
>
> We have provide more evaluations on various training splits, segmentation tasks, and have included error bars from multiple runs. These can be found below.
>
>
> >A smaller weakness of this paper is that although the authors show that equivariance is not necessary for performance, there is no explanation or hypothesis as to why this may be true. This would greatly strengthen the paper in my opinion.
>
> We would like to refrain from being too speculative. In the Discussion, we do hypothesize this may be because the small number of tokens, so it may not hold for language. However, there has been evidence in the past that equivariance as a hard constraint is too strict for vision, specifically relaxed weight sharing has beneficial for vision in [6].
>
>
> [6] Elsayed, G., Ramachandran, P., Shlens, J., & Kornblith, S. (2020, November). **Revisiting spatial invariance with low-rank local connectivity.** In International Conference on Machine Learning (pp. 2868-2879). PMLR.
>
> ---
>
> # Blanket Statement
>
> ----
> ## Significance
> As we feel like several reviewers may have misunderstood the goal of our work, we would like to clarify on this point: Our goal is understand why RoPE works, and specifically whether equivariance is the dominant contributer as is commonly believed. Our claim is *not* that one should use Spherical RoPE or Uniform RoPE. These are ablations to move towards understanding what is important for RoPE's success.
>
> Because RoPE has been incorporated into every large model, it is a very relevant question for this year to understand why exactly it works. It is strongly believed that the improvement is because RoPE is a relative positional encoding. We believe we clearly show that this is strictly not true. Whether Liere, Mixed RoPE, or Spherical RoPE perform *slightly* better is not relevant as the understanding is RoPE is better because it is equivariant. As Spherical RoPE (which is not relative) maintains its gap over APE where Uniform and Axial RoPE (which are relative) has a weakened performance, equivariance does not seem *necessary* for the main benefits of RoPE.
>
> We believe that this is significant in context of *very* recent work. Many contemporary works seek to impose constraints, such as commutativity, to "extend" LieRE. We show by Proposition 2 that if we need to create a rotary embedding which is commutative, then whatever we do will theoretically boil down to Mixed RoPE. We also show through experiments that reversibility [3] (Axial RoPE) produces worse performance. To illustrate the impact, we point the reviewers to three recent papers doing exactly this that have been accepted or submitted in the last three months:
>
> [1] Connor Schenck, Isaac Reid, Mithun George Jacob, Alex Bewley, Joshua Ainslie, David Rendleman, Deepali Jain, Mohit Sharma, Avinava Dubey, Ayzaan Wahid, et al.
> **Learning the ropes: Better 2d and 3d position encodings with string.** (ICML July 2025 Spotlight Poster)
>
> [2] Yu, H., Jiang, T., Jia, S., Yan, S., Liu, S., Qian, H., ... & Yuan, C. (2025). **ComRoPE: Scalable and Robust Rotary Position Embedding Parameterized by Trainable Commuting Angle Matrices.** In Proceedings of the Computer Vision and Pattern Recognition Conference (pp. 4508-4517).
>
> [3] Liu, H., & Zhou, H. (2025). **Rethinking RoPE: A Mathematical Blueprint for N-dimensional Positional Encoding.** (Preprint)
>
> ---
> ## Additional Experiments
>
> We will add results when they are available.
>
> We improved the frequency initialization for learnable RoPEs and have run these additional evaluations:
>
> |  Accuracy with STDev | Spherical (*Learned*)   | Axial  (*Learned*)   | Mixed | Uniform | APE| LieRE|
> | --------           | --------     | -------- | -------- |-------- | -------- | -------- |
> | CIFAR100 (5 Runs)  | 73.2 +- .4 (*74.1 +- .4*)| 72.1+-0.6 (*72.9 +- .6*)|**74.7+-.3**| 70.5+-.2 |64.2 +- 0.9|74.2
> |ImageNet (In Progress) |||
>
> | Segmentation (IoU) | Spherical | Axial (Learned)    | Mixed | Uniform | APE |
> | --------           | --------     | -------- | -------- |-------- |-------- |
> | VOC  (No Augmentation)              |     .45     | .42     | .41|
> |VOC (Simple Augmentation)|.50|.46 (.47)|.50|.45||
>
> | Portion of CIFAR100 | Spherical (Learned) | Axial (Learned) | Mixed | Uniform | APE |
> | -------- | -------- | --------   |------ | ------ |------ |
> | .2     | 56.04 (**57.2**)    | 55.3 (56.6)       | 56.9 | 52.82 |45.9
> | .4     | 63.6  **(65.34)**   |   63.3 (62.5) | 64.4 | 59.7 | 53.4
> | .6     | 67.6  (69.8)   |   66.0 (66.78)    | **70.0** | 64.1| 57.7
> | .8     | 69.8     | 69.9     |**71.6**| |59.0|
>
> ----
>
> Below we show the time it took for each method to run 100 forward passes to compare relative speed. The experiment was performed by running a dummy input of dimension (B=256, C=3, H=224, W=224) 100 times with a ViT backbone on one A100 gpu. This is simulated training time, so the rotation matrices were recalculated with every pass.
>
> | Time comparison | Spherical (Learned) | Axial (Learned)| Mixed | LieRE| APE | Uniform |
> | -------- | -------- | -------- | -------- |-------- |-------- |-------- |
> | Without torch.autocast     | 16.6s  (16.6s)| 16.5s (16.7s)   | 15.7s  | 27.4s| 13.1s | 16.5s
> |With torch.autocast| 6.7s (5.8s)|6.5s (5.7s)| 5.2s | 13.6s| 3.9s |6.6s

---

### Official Review · Reviewer_ZRjm · 2025-07-02

**Clarity:** 3
**Significance:** 2
**Originality:** 2
**Rating:** 4
**Confidence:** 2

**Summary:**

The paper explores strategies for extending RoPE (rotary positional embeddings) for application in images. Rather than using standard positional embeddings that embed token position using addition, RoPE does it by rotating the token. However, in the context of image encoders, we need to embed a 2D location rather than a 1D position which is standard. There have been various RoPE variations that have been proposed here including adding the rotation matrices, concatenating them, and using different frequencies to embed different dimensions. This paper explores two new RoPE variants: spherical rope and uniform frequency RoPE. These new variations are compared to the previously proposed methods in the context of some standard image classification tasks. The results are nuanced.

**Questions:**

none

**Ethical Concerns:**

["NO or VERY MINOR ethics concerns only"]

**Limitations:**

See my comments in strengths and weaknesses. I am most concerned that the paper does not make the right comparisons and does not zero in on the aspects of the rotary embedding that is most important.

**Quality:**

3

**Strengths And Weaknesses:**

Strengths:

-- I thought the paper was relatively easy to understand. I liked the development of RoPE and the various rotary position embeddings considered.

-- I think researchers who are interested in rotary positional embeddings will find this work useful, although the conclusions are nuanced and not clear cut.


Weaknesses:

-- The experiments (image classification) seem appropriate, but there is not clear outperformance of the proposed methods.

-- It is not clear to me how important these results are for classification performance. The results seem relevant and interesting, but in many cases the differences in performance seem small, especially on ImageNet. It might be nice to find evaluation domains/benchmarks that illustrate these differences more precisely. In particular, it would be interesting to know if there are particular types of (image) classification problems for which these methods are particularly relevant.

-- Related to the point above, I wonder if it might make sense to create some ablation studies here that could help pinpoint which aspects of the rotary embedding are most helpful.

-- It struck me odd that Prop 1 and 2 did not have formal proofs. I would convert the "explanation" text into a proof or at least include the proof in the appendix (if I missed it in the appendix, please ignore this comment).

---

> ### Author Rebuttal · Authors · 2025-07-31
>
> # Organization
>
> Because there seemed to be consistent confusion about the goal and significance about our work we have included a blanket statement at the end, after the individual responses. We have also included the additional results in that section as it was a consistent concern across reviews. We will provide the remaining numbers as soon as possible (expect Saturday for ImageNet).
>
> -----
> **Weaknesses:**
>
> >The experiments (image classification) seem appropriate, but there is not clear outperformance of the proposed methods.
>
> Our point is not to outperform, but rather to ablate whether relativity is necessary.
>
> >It is not clear to me how important these results are for classification performance. The results seem relevant and interesting, but in many cases the differences in performance seem small, especially on ImageNet. It might be nice to find evaluation domains/benchmarks that illustrate these differences more precisely. In particular, it would be interesting to know if there are particular types of (image) classification problems for which these methods are particularly relevant.
>
> We have included a segmentation benchmark in the appendix. Once again, our point is to ablate whether relative position is important and not to show dominance of Spherical RoPE. In our view, ImageNet and image classification are biased *toward* relative PEs as the translations equivariance of convolutions have proved powerful inductive biases in the past. Thus, the comparable or better performance illustrates our point more than an anti-equivariance tailored datasets. These would not provide evidence that equivariance is not a powerful inductive bias for vision. In addition, image classification is the most common benchmark for comparing 2D RoPE methods in the closely related works of Mixed-RoPE[5], LieRE[4], ComRoPE [2] and STRING[1].
>
>
>
> > Related to the point above, I wonder if it might make sense to create some ablation studies here that could help pinpoint which aspects of the rotary embedding are most helpful
>
> There seems to be a misunderstanding. Spherical RoPE and Uniform RoPE are meant to be ablations to "pinpoint which aspects of the rotary embedding are most helpful". We have modified the tables from "(Ours)->(Our Ablation)" to make that more clear and to emphasize Spherical RoPE not necessarily as a practical method, but as an ablation.
>
>
> > It struck me odd that Prop 1 and 2 did not have formal proofs. I would convert the "explanation" text into a proof or at least include the proof in the appendix (if I missed it in the appendix, please ignore this comment).
>
> Thanks for calling it out. We will add more formal proofs to the appendix.
>
>
> Limitations:
> > See my comments in strengths and weaknesses. I am most concerned that the paper does not make the right comparisons and does not zero in on the aspects of the rotary embedding that is most important.
>
> If this is not resolved please clarify and I will try to respond.
>
>
> ----
>
> # Blanket Statement
>
> ----
> ## Significance
> As we feel like several reviewers may have misunderstood the goal of our work, we would like to clarify on this point: Our goal is understand why RoPE works, and specifically whether equivariance is the dominant contributer as is commonly believed. Our claim is *not* that one should use Spherical RoPE or Uniform RoPE. These are ablations to move towards understanding what is important for RoPE's success.
>
> Because RoPE has been incorporated into every large model, it is a very relevant question for this year to understand why exactly it works. It is strongly believed that the improvement is because RoPE is a relative positional encoding. We believe we clearly show that this is strictly not true. Whether Liere, Mixed RoPE, or Spherical RoPE perform *slightly* better is not relevant as the understanding is RoPE is better because it is equivariant. As Spherical RoPE (which is not relative) maintains its gap over APE where Uniform and Axial RoPE (which are relative) has a weakened performance, equivariance does not seem *necessary* for the main benefits of RoPE.
>
> We believe that this is significant in context of *very* recent work. Many contemporary works seek to impose constraints, such as commutativity, to "extend" LieRE. We show by Proposition 2 that if we need to create a rotary embedding which is commutative, then whatever we do will theoretically boil down to Mixed RoPE. We also show through experiments that reversibility [3] (Axial RoPE) produces worse performance. To illustrate the impact, we point the reviewers to three recent papers doing exactly this that have been accepted or submitted in the last three months:
>
> [1] Connor Schenck, Isaac Reid, Mithun George Jacob, Alex Bewley, Joshua Ainslie, David Rendleman, Deepali Jain, Mohit Sharma, Avinava Dubey, Ayzaan Wahid, et al.
> **Learning the ropes: Better 2d and 3d position encodings with string.** (ICML July 2025 Spotlight Poster)
>
> [2] Yu, H., Jiang, T., Jia, S., Yan, S., Liu, S., Qian, H., ... & Yuan, C. (2025). **ComRoPE: Scalable and Robust Rotary Position Embedding Parameterized by Trainable Commuting Angle Matrices.** In Proceedings of the Computer Vision and Pattern Recognition Conference (pp. 4508-4517).
>
> [3] Liu, H., & Zhou, H. (2025). **Rethinking RoPE: A Mathematical Blueprint for N-dimensional Positional Encoding.** (Preprint)
>
> -----
>
> ## Additional Experiments
>
> We will add results when they are available.
>
> We improved the frequency initialization for learnable RoPEs and have run these additional evaluations:
>
> |  Accuracy with STDev | Spherical (*Learned*)   | Axial  (*Learned*)   | Mixed | Uniform | APE| LieRE|
> | --------           | --------     | -------- | -------- |-------- | -------- | -------- |
> | CIFAR100 (5 Runs)  | 73.2 +- .4 (*74.1 +- .4*)| 72.1+-0.6 (*72.9 +- .6*)|**74.7+-.3**| 70.5+-.2 |64.2 +- 0.9|74.2
> |ImageNet (In Progress) |||
>
> | Segmentation (IoU) | Spherical | Axial (Learned)    | Mixed | Uniform | APE |
> | --------           | --------     | -------- | -------- |-------- |-------- |
> | VOC  (No Augmentation)              |     .45     | .42     | .41|
> |VOC (Simple Augmentation)|.50|.46 (.47)|.50|.45||
>
> | Portion of CIFAR100 | Spherical (Learned) | Axial (Learned) | Mixed | Uniform | APE |
> | -------- | -------- | --------   |------ | ------ |------ |
> | .2     | 56.04 (**57.2**)    | 55.3 (56.6)       | 56.9 | 52.82 |45.9
> | .4     | 63.6  **(65.34)**   |   63.3 (62.5) | 64.4 | 59.7 | 53.4
> | .6     | 67.6  (69.8)   |   66.0 (66.78)    | **70.0** | 64.1| 57.7
> | .8     | 69.8     | 69.9     |**71.6**| |59.0|
>
> ----
>
> Below we show the time it took for each method to run 100 forward passes to compare relative speed. The experiment was performed by running a dummy input of dimension (B=256, C=3, H=224, W=224) 100 times with a ViT backbone on one A100 gpu. This is simulated training time, so the rotation matrices were recalculated with every pass.
>
> | Time comparison | Spherical (Learned) | Axial (Learned)| Mixed | LieRE| APE | Uniform |
> | -------- | -------- | -------- | -------- |-------- |-------- |-------- |
> | Without torch.autocast     | 16.6s  (16.6s)| 16.5s (16.7s)   | 15.7s  | 27.4s| 13.1s | 16.5s
> |With torch.autocast| 6.7s (5.8s)|6.5s (5.7s)| 5.2s | 13.6s| 3.9s |6.6s

---

### Official Review · Reviewer_HWL4 · 2025-07-03

**Clarity:** 4
**Significance:** 3
**Originality:** 4
**Rating:** 4
**Confidence:** 2

**Summary:**

The core subject of this paper is positional encoding in two-dimensional data (vision), in particular, whether relative positional encodings are beneficial in this setting. The paper proved the Lie RoPE can be expressed with learned Mixed RoPE, solving an ambiguity in research. They also provide sufficient conditions for Lie RoPE to be equivariant. They then introduce Spherical RoPE, a generalization of Mixed RoPE to frquencies in 3D space, which is non-equivariant. Experimentally, they observe that equivariant approaches such as Axial RoPE do not hold an advantage.

**Questions:**

Please address the weakness section. At large, I still don't fully understand the following:
1. Can't we look at experiments without the Spherical RoPE, as it is just an example of a RoPE that isn't equivariant, while Lie RoPE is already a known one that isn't equivariant?
2. The experiments don't show IMHO that equivariance is not beneficial, it seems Axial RoPE performs better with fewer parameters (or rotational degrees of freedom).
3. Empirical evaluation is quite limited. Are there any more experiments performed? (Suggestion: Lie RoPE with commuting generators vs arbitrary generators. This might show the inverse result, that equivariance might be beneficial.)

**Ethical Concerns:**

["NO or VERY MINOR ethics concerns only"]

**Final Justification:**

The additional experiments and running time ablations show me that even if there is no positional equivariance, positional encoding learn just as well with similar running time. Due to the paper's empirical results and theoretical grounding in that Lie generators can act as a particular case of equivariant methods when their generators commute, I update my score. I do acknowledge that I understand less than what I thought about this field.

**Quality:**

3

**Strengths And Weaknesses:**

Strengths:
1. Spherical RoPE is a strong generalization essentially from 2D space to 3D allowing for more refined rotational schemes
2. Mathematically coherent argument of the expressivity of Lie vs. Learned Mixed RoPE.
3. Experiments justifying claims
Weaknesses:
1. Lie RoPE isn't equivariant if the generators don't commute, making the spherical RoPE redundant in terms of figuring out whether equivariance is important (we can just compare Lie RoPE to itself with/without commutative generators)
2. Experiments are shown without standard deviation; thus results aren't clear, the results themselves are quite close to each other (other than in comparison to APE).
3. Axial RoPE (which is equivariant) performs consistently better or at the same level as Spherical with fewer degrees of freedom and parameters, so it may indicate that equivariance is important...

---

> ### Author Rebuttal · Authors · 2025-07-31
>
> # Organization
>
> Because there seemed to be consistent confusion about the goal and significance about our work we have included a blanket statement at the end, after the individual responses. We have also included the additional results in that section as it was a consistent concern across reviews, we will provide the remaining numbers as soon as possible (expect Saturday for ImageNet).
>
> ------
>
> **Weaknesses:**
> >Lie RoPE isn't equivariant if the generators don't commute, making the spherical RoPE redundant in terms of figuring out whether equivariance is important (we can just compare Lie RoPE to itself with/without commutative generators)
>
> By Proposition 2, all LieRE with commutative generators are theoretically equivalent to some form of Mixed RoPE, as we show it is the spectral decomposition of the generators. Thus, to compare "unconstrained" LieRE to LieRE with commutative generators, we can theoretically compare the numbers to Mixed RoPE which is done in their [4] or add positional embeddings as done in [5].
>
> However, these do not isolate equivariance. Spherical RoPE has three main advantages as an ablation:
>
>
> - **LieRE is capable of learning commutative generators where (fixed) Spherical RoPE is strictly unable to (in one attention layer).** Similarly, modifying Mixed RoPE with APE as done in the Mixed RoPE paper (Heo et al.(2024)) makes the model not equivariant, however it is not a precise ablation. In a similar way to how Mixed RoPE constrains to LieRE to be commutative generators, Spherical is constrained to be non-commutative generators -- outside of the degenerate case where the frequencies are learned to be zero. This is why we choose it to evaluate whether equivariance is why RoPE performs well. In summary, the important property is **Spherical RoPE is strictly the component of LieRE which breaks the "relative positional encoding".**
> - **LieRE does not have as comparable of an implementation**. LieRE requires exponentiating skew symmetric matrices which: 1. Requires more parameters 2. Introduces numerical caveats. **Spherical RoPE on the other hand can be implemented in an almost identical way to Axial and Mixed RoPE making it more appropriate as an ablation.**
> - **The learned behavior of Spherical/Mixed/Axial RoPE is more concise.** We will add a section visualizing the weights learned to the appendix in each layer by Spherical, Axial and Mixed RoPE. We observe that that they learn to set many frequencies to zero. This also means weight decay on the frequencies will bias the model to be position invariant. This type of intuition is more difficult when learning Lie matrices with LieRE or other commutative variants.
>
> [4] Ostmeier, S., Axelrod, B., Moseley, M. E., Chaudhari, A., & Langlotz, C. (2024). Liere: Generalizing rotary position encodings. arXiv preprint arXiv:2406.10322.
>
> [5] Heo, B., Park, S., Han, D., & Yun, S. (2024, September). Rotary position embedding for vision transformer. In European Conference on Computer Vision (pp. 289-305). Cham: Springer Nature Switzerland.
>
>
>
> >Experiments are shown without standard deviation; thus results aren't clear, the results themselves are quite close to each other (other than in comparison to APE).
>
> This is one of the main points. Even completely removing equivariance by constraining the generators to be non-commutative the results remain close outside of Uniform RoPE and APE. We have improved results for learned PEs and included error bars from multiple runs (see above evaluations).
>
> >Axial RoPE (which is equivariant) performs consistently better or at the same level as Spherical with fewer degrees of freedom and parameters, so it may indicate that equivariance is important...
>
> Spherical RoPE outperforms Axial RoPE, neither of which require any parameters. Here is the table for training on subsamples. Spherical RoPE performs consistently better.
>
> | Portion of CIFAR100 | Spherical | Axial |
> | -------- | -------- | -------- |
> | .2     | **56.04**     | 55.3     |
> | .4     | **63.6**     | 	63.3     |
> | .6     | **67.6**     |	66.0     |
> | .8     | 69.8     | **69.9**     |
> | Full     | **73.2(+-.4)**     | 72.1+-.6     |
> | Added Parameters     | 0     | 0    |
>
>
> **Questions:**
> >Can't we look at experiments without the Spherical RoPE, as it is just an example of a RoPE that isn't equivariant, while Lie RoPE is already a known one that isn't equivariant?
>
> **See response to weaknesses**
>
> >The experiments don't show IMHO that equivariance is not beneficial, it seems Axial RoPE performs better with fewer parameters (or rotational degrees of freedom).
>
> **See response to weaknesses**
>
> >Empirical evaluation is quite limited. Are there any more experiments performed? (Suggestion: Lie RoPE with commuting generators vs arbitrary generators. This might show the inverse result, that equivariance might be beneficial.)
>
> In our paper, we show that Lie RoPE with commuting generators is the same (mathematically) as Mixed RoPE, so your suggestion boils down to comparing their performances. See first response for more detail.
>
>
> # Blanket Statement
>
> ----
> ## Significance
> As we feel like several reviewers may have misunderstood the goal of our work, we would like to clarify on this point: Our goal is understand why RoPE works, and specifically whether equivariance is the dominant contributer as is commonly believed. Our claim is *not* that one should use Spherical RoPE or Uniform RoPE. These are ablations to move towards understanding what is important for RoPE's success.
>
> Because RoPE has been incorporated into every large model, it is a very relevant question for this year to understand why exactly it works. It is strongly believed that the improvement is because RoPE is a relative positional encoding. We believe we clearly show that this is strictly not true. Whether Liere, Mixed RoPE, or Spherical RoPE perform *slightly* better is not relevant as the understanding is RoPE is better because it is equivariant. As Spherical RoPE (which is not relative) maintains its gap over APE where Uniform and Axial RoPE (which are relative) has a weakened performance, equivariance does not seem *necessary* for the main benefits of RoPE.
>
> We believe that this is significant in context of *very* recent work. Many contemporary works seek to impose constraints, such as commutativity, to "extend" LieRE. We show by Proposition 2 that if we need to create a rotary embedding which is commutative, then whatever we do will theoretically boil down to Mixed RoPE. We also show through experiments that reversibility [3] (Axial RoPE) produces worse performance. To illustrate the impact, we point the reviewers to three recent papers doing exactly this that have been accepted or submitted in the last three months:
>
> [1] Connor Schenck, Isaac Reid, Mithun George Jacob, Alex Bewley, Joshua Ainslie, David Rendleman, Deepali Jain, Mohit Sharma, Avinava Dubey, Ayzaan Wahid, et al.
> **Learning the ropes: Better 2d and 3d position encodings with string.** (ICML July 2025 Spotlight Poster)
>
> [2] Yu, H., Jiang, T., Jia, S., Yan, S., Liu, S., Qian, H., ... & Yuan, C. (2025). **ComRoPE: Scalable and Robust Rotary Position Embedding Parameterized by Trainable Commuting Angle Matrices.** In Proceedings of the Computer Vision and Pattern Recognition Conference (pp. 4508-4517).
>
> [3] Liu, H., & Zhou, H. (2025). **Rethinking RoPE: A Mathematical Blueprint for N-dimensional Positional Encoding.** (Preprint)
>
> ---
> ## Additional Experiments
>
> We will add results when they are available.
>
> We improved the frequency initialization for learnable RoPEs and have run these additional evaluations:
>
> |  Accuracy with STDev | Spherical <br>*Learned*   | Axial <br> *Learned*   | Mixed | Uniform | APE| LieRE|
> | --------           | --------     | -------- | -------- |-------- | -------- | -------- |
> | CIFAR100 (5 Runs)  | 73.2 +- .4 (*74.1 +- .4*)| 72.1+-0.6 (*72.9 +- .6*)|**74.7+-.3**| 70.5+-.2 |64.2 +- 0.9|74.2
> |ImageNet (In Progress) |||
>
> | Segmentation (IoU) | Spherical | Axial (*Learned*)    | Mixed | Uniform | APE |
> | --------           | --------     | -------- | -------- |-------- |-------- |
> | VOC  (No Augmentation)              |     .45     | .42     | .41|
> |VOC (Simple Augmentation)|.50|.46 (*.47*)|.50|.45||
>
> | Portion of CIFAR100 | Spherical (*Learned*) | Axial (*Learned*) | Mixed | Uniform | APE |
> | -------- | -------- | --------   |------ | ------ |------ |
> | .2     | 56.04 (**57.2**)    | 55.3 (*56.6*)       | 56.9 | 52.82 |45.9
> | .4     | 63.6  **(65.34)**   |   63.3 (*62.5*) | 64.4 | 59.7 | 53.4
> | .6     | 67.6  (*69.8*)   |   66.0 (*66.78*)    | **70.0** | 64.1| 57.7
> | .8     | 69.8     | 69.9     |**71.6**| |59.0|
>
> ----
>
> Below we show the time it took for each method to run 100 forward passes to compare relative speed. The experiment was performed by running a dummy input of dimension (B=256, C=3, H=224, W=224) 100 times with a ViT backbone on one A100 gpu. This is simulated training time, so the rotation matrices were recalculated with every pass.
>
> | Time comparison | Spherical (Learned) | Axial (Learned)| Mixed | LieRE| APE | Uniform |
> | -------- | -------- | -------- | -------- |-------- |-------- |-------- |
> | Without torch.autocast     | 16.6s (16.6s)| 16.5s (16.7s)   | 15.7s  | 27.4s| 13.1s | 16.5s
> |With torch.autocast| 6.7s (5.8s)|6.5s (5.7s)| 5.2s | 13.6s| 3.9s |6.6s

---

> > ### Comment · Reviewer_HWL4 · 2025-08-05
> > **Thanks**
> >
> > Thank you for addressing my main concerns. The additional experiments and running time ablations show me that even if there is no positional equivariance, positional encoding learn just as well with similar running time. Due to the paper's empirical results and  theoretical grounding in that Lie generators can act as a particular case of equivariant methods when their generators commute, I update my score.

---

### Note · Authors · 2025-08-13

We have addresses the reviewers' original concerns by several control experiments. This additional evaluation greatly strengthened our paper. For instance, we now show that our findings hold even in low data regimes, where equivariance would intuitively be favored. We also show similar behavior for segmentation and added error bars to indicate the consistency of our findings.

In addition, as Reviewer 5p81 suggested, we added nuance more directly in our wording rather than leaving it for the reader. We believe this also addresses the misunderstandings with Reviewer Kc7V.

While the remaining suggestions of the reviewers are interesting, they are somewhat tangential to our main claim: We show the belief that "RoPE performs well because it is a relative positional encoding" is unjustified. Given the recent strong interest in positional encodings, this is an important finding as it goes against commonly held belief.

---

### Decision · Program_Chairs · 2025-09-17

**Decision:**

Accept (poster)

**Comment:**

The main concern initially raised by a subset of reviewers was on the main finding of the paper; after an engaging discussion round, it seems there is now understanding that the main point of the paper was not to promote the advantages of Spherical RoPE or Uniform RoPE, but to perform ablation studies relative to these two methods to better understand why RoPE works. All reviewers find value in the findings reported in the paper. Although one reviewer believes additional and more comprehensive study is needed to make the argument that equivariance is not the primary reason for RoPE's success, the study itself and its results are still considered meaningful and of value. The final version of the paper will, I believe, be considerably revised to reflect the many constructive suggestions of the reviewers, adding more nuance to the claims and better explaining the main findings about the relationship between equivariance and RoPE's success.